# Long-term hazard of pyroclastic density currents at Vesuvius (Southern Italy) with maps of impact parameters

Pierfrancesco Dellino[1], Fabio Dioguardi[1], Roberto Sulpizio[1,2,3], Daniela Mele[1]

[1]Dipartimento di Scienze della Terra e Geoambientali, Università degli Studi di Bari "Aldo Moro", Bari, 70125, Italy
[2] Istituto Nazionale di Geofisica e Vulcanologia, Bologna section, Italy
[3] IGAG-CNR, Milano, Italy

*Correspondence to*: Pierfrancesco Dellino (pierfrancesco.dellino@uniba.it)

**Abstract.** The hazard of pyroclastic density currents (PDCs) at Vesuvius is investigated by analysing deposits from past eruptions. No specific eruption was chosen as representative of the hazard of PDCs, and the analysis is extended to all the eruptions that left substantial deposits on the ground. Based on the stratigraphic evidence, we assume that at Vesuvius the currents are bipartite, with a basal highly-concentrated part, which was fed from the collapse of the eruptive fountain on the ground, and an overlying part generated by the expulsion of gas and fines that fed a dilute and turbulent shear flow. Dynamic pressure, particle volumetric concentration, temperature and flow duration are hazardous characteristics of PDCs that can impact buildings and population and are defined here as impact parameters. They have been calculated by means of an implementation of the PYFLOW code, which uses the deposit particle characteristics as input. The software searches for the probability density function of impact parameters. The 84th percentile has been chosen as a safety value of the expected impact at long term (50 years). There is no correlation between eruption size and impact parameters. Maps have been constructed by interpolation of the safety values calculated at various points over the dispersal area, and show how impact parameters change as a function of distance from the volcano. The maps are compared with the red zone, which is the area that the National Department of the Italian Civil Protection has declared to be evacuated under conditions of an impending eruption. The damaging capacity of currents over buildings and population is discussed both for the highly concentrated part and the diluted one.

## 1 Introduction

Pyroclastic density currents (PDCs) originate from a variety of processes during explosive volcanic eruptions, e.g. the fountaining of the gas-particle mixture (aka eruption column) issuing from a crater or the avalanching of a volcanic dome. In the first case, the parent current can evolve into a highly concentrated, poorly sorted underflow and an overlying, dilute, fully-turbulent current (Sulpizio et al., 2014). PDCs represent the most hazardous events of volcanic eruptions, with historic cases causing destruction and deaths over large areas (Baxter et al., 1998; Cao et al., 2003; Sulpizio et al., 2014). Understanding the

processes characterizing PDCs, such as transport and deposition of pyroclastic particles, from the study of deposits emplaced by PDCs is essential for developing effective hazard assessment and risk management strategies (Jones et al. 2023).

Various attempts have been made to define specific flow characteristics that are useful for evaluating the damaging capacity of PDCs, such as dynamic pressure, which is a measure of the impact force per unit area of a current that can exert lateral loads onto buildings (Valentine et al., 1998; Spence et al., 2004; Zuccaro et al., 2008). Other damaging factors, which we define

here as impact parameters, are the flow temperature, the content of ash particles, projectiles carried by the current and the duration of the flow, which directly or indirectly affect the survivability of people caught unprotected by a PDC (Horwell and Baxter, 2006; Jenkins et al., 2013; Baxter et al., 2017). The latter become important especially over distal areas, where the strength of the current decays but the lethal effect of the gas-particle mixture remains, as occurred at Pompeii during the historical eruption of 79 AD (Dellino et al., 2021), which represents an invaluable source of information of the actual impact

of PDCs.

No systematic analysis of these flow characteristics as deduced from deposit properties has been made so far for assessing quantitatively the PDCs' hazard around a volcano. At Vesuvius, PDC deposits have been studied in previous papers (Cioni et al., 2004; Neri et al. 2007; Sulpizio et al., 2007; Esposti Ongaro et al. 2008; Dellino et al., 2008; Zanella et al. 2008; Sulpizio et al., 2010a; Gurioli et al., 2010; Mele et al., 2011; Zanella et al. 2015;  Giordano et al., 2018; Dellino et al., 2021; Pensa et

al., 2023), but a detailed investigation of the impact parameters for the aim of a probabilistic hazard assessment is still not available. PDCs at Vesuvius represent a significant source of risk, being the area surrounding the volcano highly populated, with around 700,000 inhabitants living in the red zone, the area to be evacuated in case of an impending eruption (Figure 1; Civil Protection Department, 2014). The impacts that the PDCs could have on buildings or people are not quantified on the red-zone map. In this paper, we try to fill this gap and investigate the distribution of the impact parameters over the volcano's

surroundings, including the red zone. PDC deposits from previous Vesuvius eruptions provide key information that can be used to deduce impact parameters from potential future eruptions. To follow this line, it is necessary to investigate the PDC deposits first, then define a general model of the current that links deposit characteristics to flow dynamics, and finally reconstruct the impact parameters that better represent flow intensity in terms of damage potential.

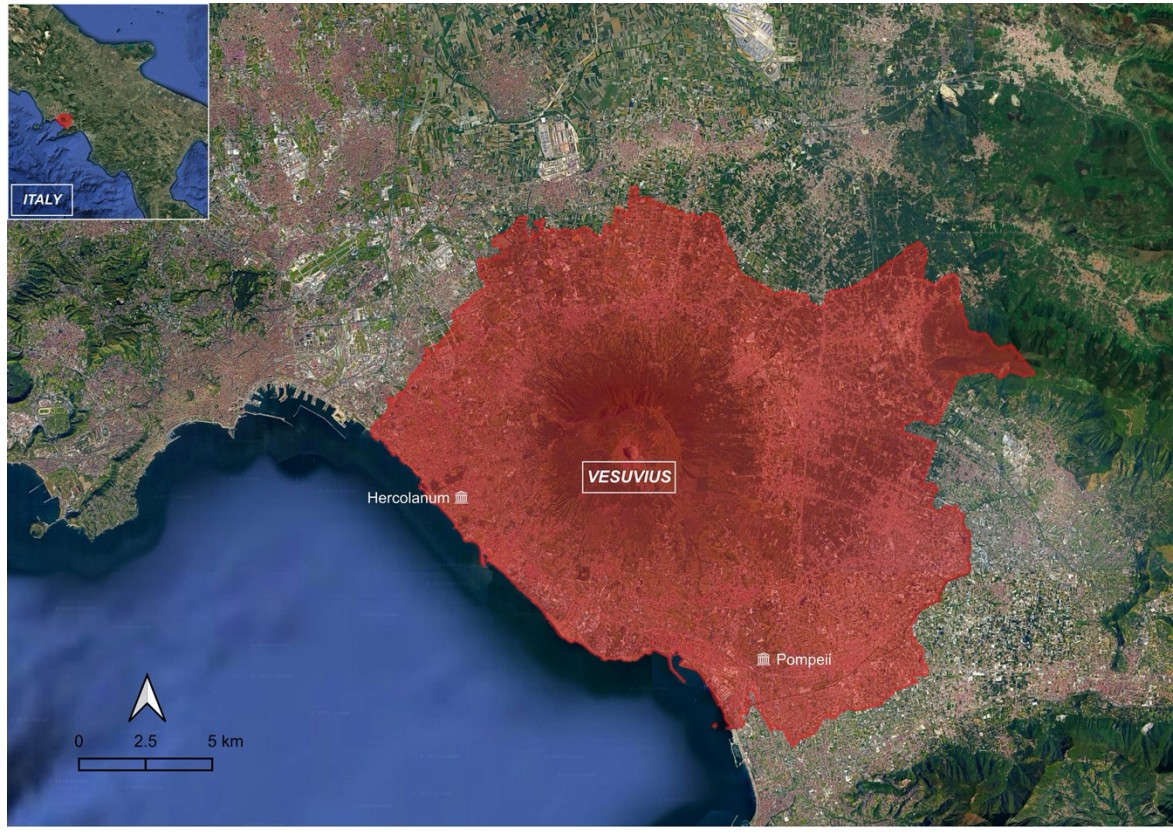


**Figure 1. The red zone of the evacuation plans for Vesuvius (from https://www.protezionecivile.gov.it/it/approfondimento/aggiornamento-del-piano-nazionale-di-protezione-civile-il-vesuvio/, last access: 19 March 2025; base map is from © Google Earth).**

## 2 Reconstruction of the facies architecture of PDC deposits


Stratigraphic evidence of volcanism at Vesuvius dates back to around 39 cal. ky BP (Brocchini et al., 2001; Santacroce et al., 2008), with predominantly effusive eruptions forming the Monte Somma volcano. About 22 cal. ky BP, the activity changed into largely explosive eruptions, which formed the summit caldera complex of Mt Somma (Cioni et al., 1999). After the Pompeii eruption's Plinian event of 79 AD, volcanism continued mainly within the Mt Somma caldera, with the construction

of modern Mount Vesuvius. The most recent eruption occurred in AD 1944 (Cole and Scarpati, 2010). The best preserved PDC deposits are from the eruptions of Pomici di Mercato (8.9 cal. ky BP; Santacroce et al., 2008; Mele et al., 2011), Pomici di Avellino (3.9 cal. ky BP; Sulpizio et al., 2010b; Sevink et al., 2011), AP2 (3.5 cal. ky BP; Cioni et al., 2008), Pompeii (AD 79; Sigurdsson et al., 1985; Cioni et al., 1992), Pollena (AD 472; Sulpizio et al., 2005) and AD 1631 (Rosi et al., 1993). PDC deposits from other eruptions can be found (i.e., Pomici di Base and Greenish; Bertagnini et al., 1998; Cioni et al., 2003), but

do not have sufficient continuity of exposure that is necessary for the hazard analysis of the present research. According to Selva et al. (2022), there is around 34% probability of an eruption at Vesuvius in the next fifty years, which we consider as a reference time for the long-term hazard. Spotty data about the intensity of PDCs at Vesuvius in terms of potential damage have been published in the past using both geological data (Sulpizio et al. 2010; Mele et al. 2011; Dellino et al. 2021) or numerical simulations (e.g. Neri et al. 2007; Esposti Ongaro et al. 2008), but a comprehensive assessment of the expected intensity, i.e.,

a quantification of the impact parameters, of PDCs in the Vesuvius area is still not available.

To take the eruptive history of Vesuvius into account and get an unbiased range of the variation of impact parameters, all the eruptions that show well-preserved deposits in the field are considered in this paper. This means that no particular event is used to propose a specific hazard scenario, but all the suitable PDC-forming eruptions are considered, in order to obtain a representative sample of the impact parameters of PDCs. It is worth noting that our approach may be biased towards the larger

PDCs, which are better represented in the geological record. However, these are the flows that can have a significant impact in the area under analysis. Our assessments are not based on the calculation of probability of occurrence, which would require a thorough knowledge of all the PDCs events. This means that the probabilistic information presented here is conditional upon an eruption occurring.

Field studies, which extended from the gullies on the volcano flank to the plain surrounding Vesuvius, show that a PDC deposit is commonly composed of a repetitive succession of beds with stratigraphic continuity. Combining observations of all deposits, a general "facies architecture" has been defined, synthesizing the lateral and vertical succession of beds associated with a current (Figure 2). The general facies architecture records the common behaviour of PDC emplacement at Vesuvius.

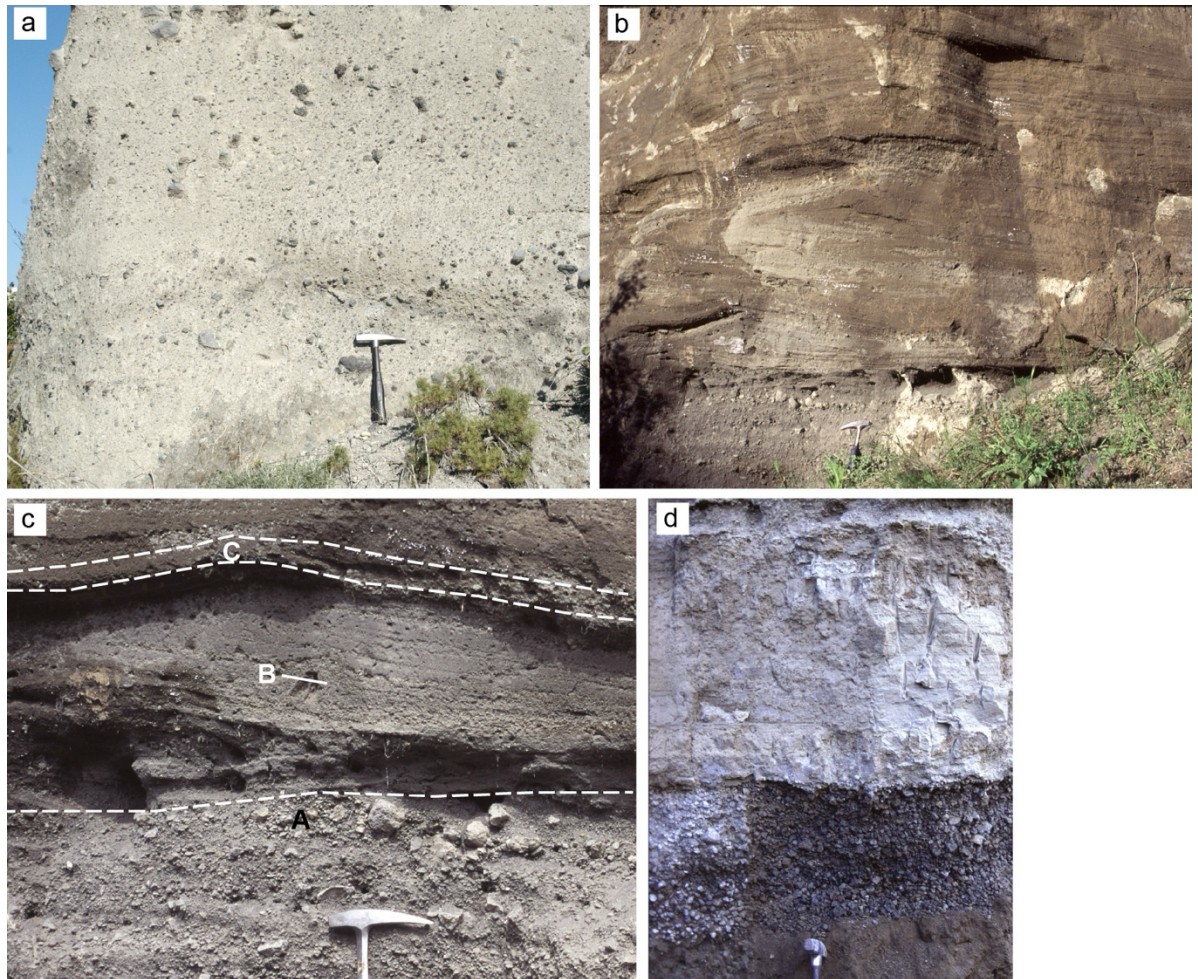

Figure 2. Deposits of pyroclastic density currents at Vesuvius. a) a massive, structureless deposit of ash, lapilli and bombs. b) a decimeter to meter thick dune-bedded layer of ash and lapilli with internal lamination and traction structures. c) the fining upward succession formed by the passage of a PDC: A = coarse clast entrained at the base of the current. B = laminated layer. C = thin, fine ash layer. The dashed lines separate the three layers. d) Fine-ash deposit (white layer).

In the proximal area, along the gullies that cut the volcano slope, the vertical facies architecture is generally composed of meters thick poorly-sorted massive layer of ash, lapilli and bombs (Figure 2a). It is overlain by a decimetre-to-meter thick stratified, sometimes dune-bedded horizon of ash and lapilli with internal lamination and traction structures (Figure1b, c). The facies architecture is capped by fine-grained ash layer(s) of centimetric thickness (Figure1c). Coarse grained massive facies occur close to the break in slope between the volcano and the surrounding apron, at the mouth of the main valleys draining the volcano slopes. Stratified facies predominate in interfluve deposits, and, for Avellino and Pompeii eruptions, this stratified facies occurs also beyond the break-in slope up to the distal area (tens of km from the volcano). As a general rule, the stratified

facies decreases in thickness and grain size with distance from the source, while the fine-grained ash facies remains almost constant in grain size and thickness (Figure1d).

In the volcanological literature, massive and stratified deposits have been interpreted either in terms of distinct flows or in terms of bipartite currents in which the massive part refers to a highly concentrated undercurrent and the stratified one to the overlying dilute turbulent current (commonly referred to as ash-clod surge). In other cases, the massive deposit has been interpreted as syn-sedimentation reconcentration and/or rapid remobilization of material first deposited by a primary, stratified current (Druitt et al., 2002; Valentine et al., 2022). In the case of Vesuvius, we interpret the deposits as the result of a bipartite current, because the massive deposit and the overlying stratified one are always in stratigraphic continuity, which means that no erosional surfaces are found in between layers, hence no temporal breaks are associated to the layers emplacement. Following this interpretation, the facies cropping out along the gullies are the result of a basal highly-concentrated underflow, which forms the massive facies, and an overlying stratified facies resulting from the dilute current and the capping fine-ash recording the settling of lingering ash after passage of the current. The contemporaneous occurrence of a massive underflow together with a dilute overcurrent has already been reported (Fisher, 1979; Cas and Wright, 1987; Gernon et al., 2013; Breard and Lube, 2017). This deposit architecture can be interpreted in terms of a current that in its early phase of development was separated into two parts, depending on a different balance between the sedimentation rate and the bedload flux (Dellino et al., 2019; 2020).

Our interpretation is that the massive layer was fed directly from the collapse of the eruptive fountain on the ground, which was characterized by a high sedimentation rate that damped turbulence due to a high particle concentration. It has already been demonstrated that thick, massive deposits can be formed because of a high sedimentation rate, which inhibits traction at the bedload (Lowe, 1982, 1988; Fisher, 1990; Druitt, 1992; Kneller and Branney, 1995; Branney and Kokelaar, 2002; Woods et al., 2002; Postma et al., 2009). Furthermore, experiments show that massive beds are formed from suspension where the sedimentation rate exceeds the bedload flux by two orders of magnitude (Dellino et al., 2010; 2019). The underflow was channelised along the volcano valleys and stopped abruptly at the break-in slope (Figure 2a).

The lateral stress generated by the collapse on the ground of the eruption fountain led to the expulsion of part of the collapsed material and fed an overlying shear flow decoupled from the massive flow (Sweeney and Valentine, 2017; Valentine and Sweeney, 2018; Dellino et al., 2020). It evolved laterally into a highly expanded, fully turbulent, gas-particle current that formed both the stratified facies (Figure 2b, c) and the fine-grained ash from gentle settling of the suspended material during the waning phase of the current. The fine ash has a sedimentation time and can be easily drifted away from the lower atmosphere winds over the plain surroundings of Vesuvius.

## 3 Physical modelling of impact parameters: the example from the Pomici di Mercato eruption

Before showing the hazard intensity maps obtained by integrating data from all eruptions, the approach used in the reconstruction of the impact parameters is illustrated by the example of one PDC deposit of the Pomici di Mercato eruption.

The stratigraphy of this eruption (Mele et al., 2011) is made up of alternating fallout and PDC deposits that are well-exposed in the northern sector of the volcano (Figure 3a). The massive bed and the overlying stratified one are in stratigraphic continuity and can be traced for a sufficient distance without showing erosional contact. Therefore, we interpret them as the result of a bipartite current. The PDC deposits considered here are from the first phase of the eruption and were generated by the collapse on the ground of an eruptive fountain, whose tentative location is represented by the black dot at the rim of Figure 3a, as deduced by the maximum thickness (2 m) of deposits. The deposit sequence consists of a meter-thick, poorly-sorted massive layer of lapilli and scattered bombs and blocks set in an ash matrix (Figure 3b, c), which is related to the highly-concentrated underflow, and a dune-bedded, stratified layer (Figure 3c, d), which is related to the overlying dilute current (Mele et al., 2011; Dellino et al., 2019). When cropping out on the gentle slope of the volcano flank, the stratified layer shows a thickness of 0.5 m and small dunes of lapilli and ash of 1 m wavelength and 0.1 m in height (Figure 3c)

The physical characteristics of the bipartite current need to be reconstructed by means of two separate models.

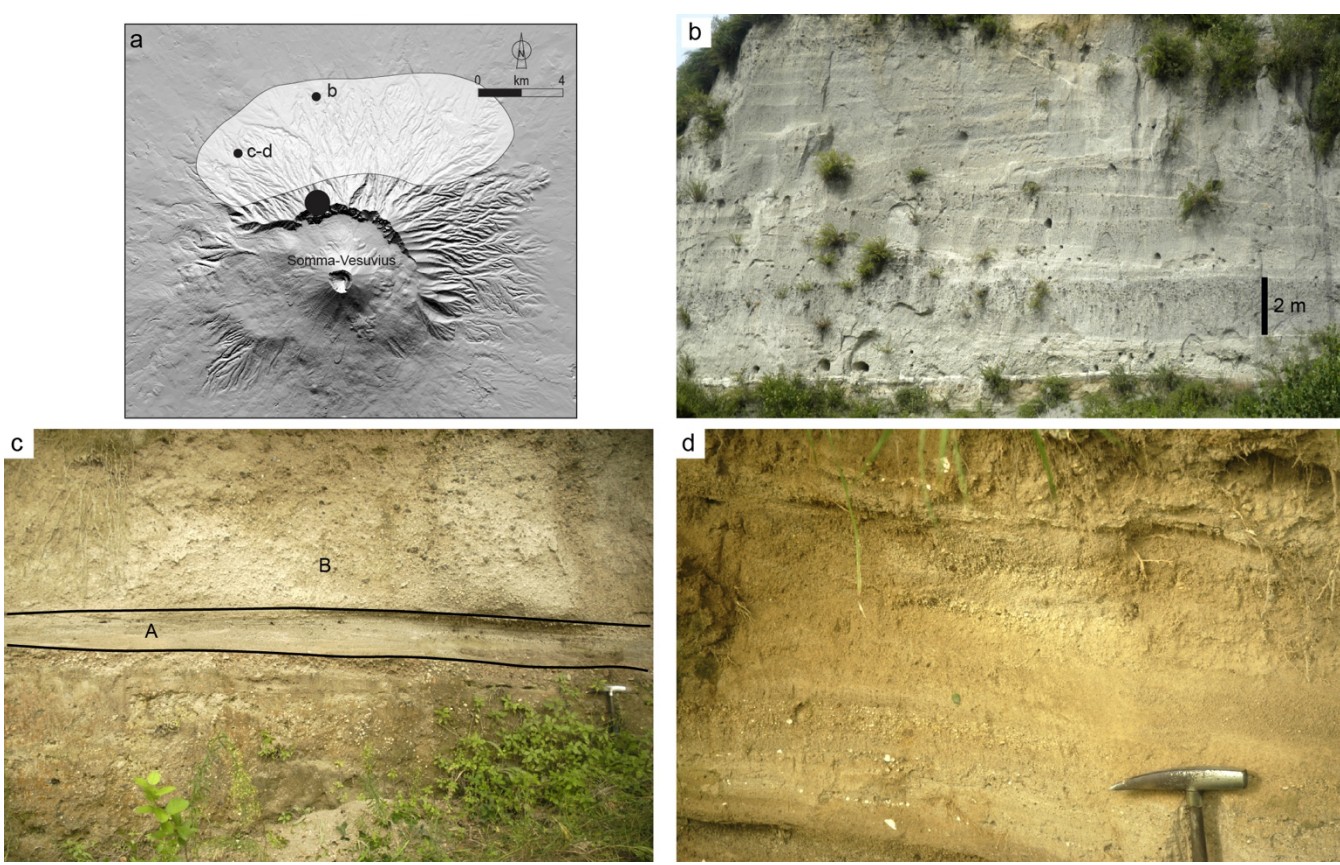

Figure 3. Field pictures showing the Pomici di Mercato eruption deposits used as a case study. (a) The labels b, c, and d refer to the deposits shown in b, c, and d. The c-d location corresponds to the MC13/4 massive and MC13/4 samples in Table 2. The black circle represents the zone of impact of the eruptive fountain (the digital elevation model by Tarquini et al., 2023). The white shaded area represents the dispersal area of the PDCs. (b) The massive deposit emplaced by freezing at the break in slope, indicated by the meter stick. (c) The stratified layer (A) that, being faster arrived before the massive layer (B) at the split location. (d) The hammer points at the stratified layer on top of the massive layer at the split location.

In the following, the model of the overlying dilute current, which represents the deposition of the dune-bedded layer, is discussed first. Afterward, the model of the highly-concentrated undercurrent forming the massive bed, is presented. This order is justified by the fact that data from the overlying dilute current provides information on the underlying concentrated one.

### 3.1 Model of the overlying dilute current forming the dune-bedded layer

The overlying dilute current, which formed the stratified dune-bedded layer, is modelled as a turbulent boundary layer shear flow (Furbish, 1997; Dellino et al., 2008) that carries solid particles into suspension. Flow movement is initiated by the gravity acting on the current along the volcano slope, which is due to the density difference between the volcanic gas-particle mixture and the surrounding atmosphere. The current is made up of a mixture of magmatic gas, volcanic particles, and air entrained by turbulence during runout. It is stratified in terms of velocity and particle concentration, hence density (Middleton and Southard, 1984; Valentine, 1987). As a consequence of sedimentation and air entrainment, the volumetric particle concentration decreases along the flow path down to a point where the density difference with atmosphere is nullified, and the current stops its lateral movement and may become buoyant. The final deposition from the buoyant part of the current forms the fine-ash layer that closes the layer sequence.

The distribution of particles of different size, density and shape in the PDC deposit, suggests that a link exists between current flow dynamics and particles that are first taken into suspension, then sedimented into a bed load, and finally moved by traction on the ground. Such links allow the use of particle characteristics (size, density and shape), as measured in the laboratory on sediment samples, for constraining the flow model and calculate the impact parameters.

A detailed formulation of the physical model (Dellino et al., 2008) and of the numerical software code PYFLOW v2.5 (Dioguardi and Dellino, 2014; Dioguardi and Mele, 2018) is deferred to Appendix A. The main data used as input are reported in the Zenodo repository (Mele et al., 2024). Here, only the main principles of the probabilistic approach are summarised.

The basic assumption is that, at sedimentation, the settling velocity of particles equals the current shear velocity (Middleton and Southard, 1984; Dellino et al., 2008), which is a quantity that, together with flow density, allows estimation of the impact parameters (dynamic pressure, particle concentration, temperature, flow duration). The settling velocity depends on the particle characteristics, mainly grain size, density and aerodynamic coefficients. Deposits are characterized by a broad distribution of particle sizes and densities, which can result from unsteady flow fluctuations that, upon sedimentation, affect shear velocity and settling velocity. In order to take such unsteadiness into account, solutions are provided in terms of the probability density function (PDF) of the grain-size distribution of deposit samples. In this paper, the solution corresponding to the 84[th] percentile of the PDF is used in the maps of impact parameters. It is considered a safety value for evaluating the damaging effect of the impact parameters. The method has been validated by large-scale experiments (Dellino et al., 2010), where values of the impact parameters measured by sensors fell well in the range of solution of the probability density function. Also, it was demonstrated that the solid particles temperature did not change much before the experiment and in the deposits after the experiment. It means that the simple model of flow temperature we used, which takes into consideration only the gas-particle average mixture

temperature and not the interphase heat exchange between phases, is an acceptable approximation of the average temperature of the gas-particle mixture. In the following, the results of flow dynamic pressure, particle volumetric concentration, temperature, and flow duration, which represent the impact parameters, are illustrated for an example of the Pomici di Mercato eruption.

### 3.1.1 Flow dynamic pressure and particle volumetric concentration

In order to illustrate how the calculated flow characteristics vary vertically in the stratified current, the profiles of particle concentration, density, velocity, and dynamic pressure calculated from a single outcrop of a Pomici di Mercato's PDC deposit are shown in Figure 4. Results are presented by means of the 84[th], 50[th] and 16[th] percentile of the PDF, which were calculated with the method of Dioguardi and Dellino (2014; see the method in Appendix A) and show the statistical variability in terms of percentiles. Here, we present the vertical profiles up to 50 m above the ground level, which is the minimum estimate of the total flow thickness calculated by PYFLOW.

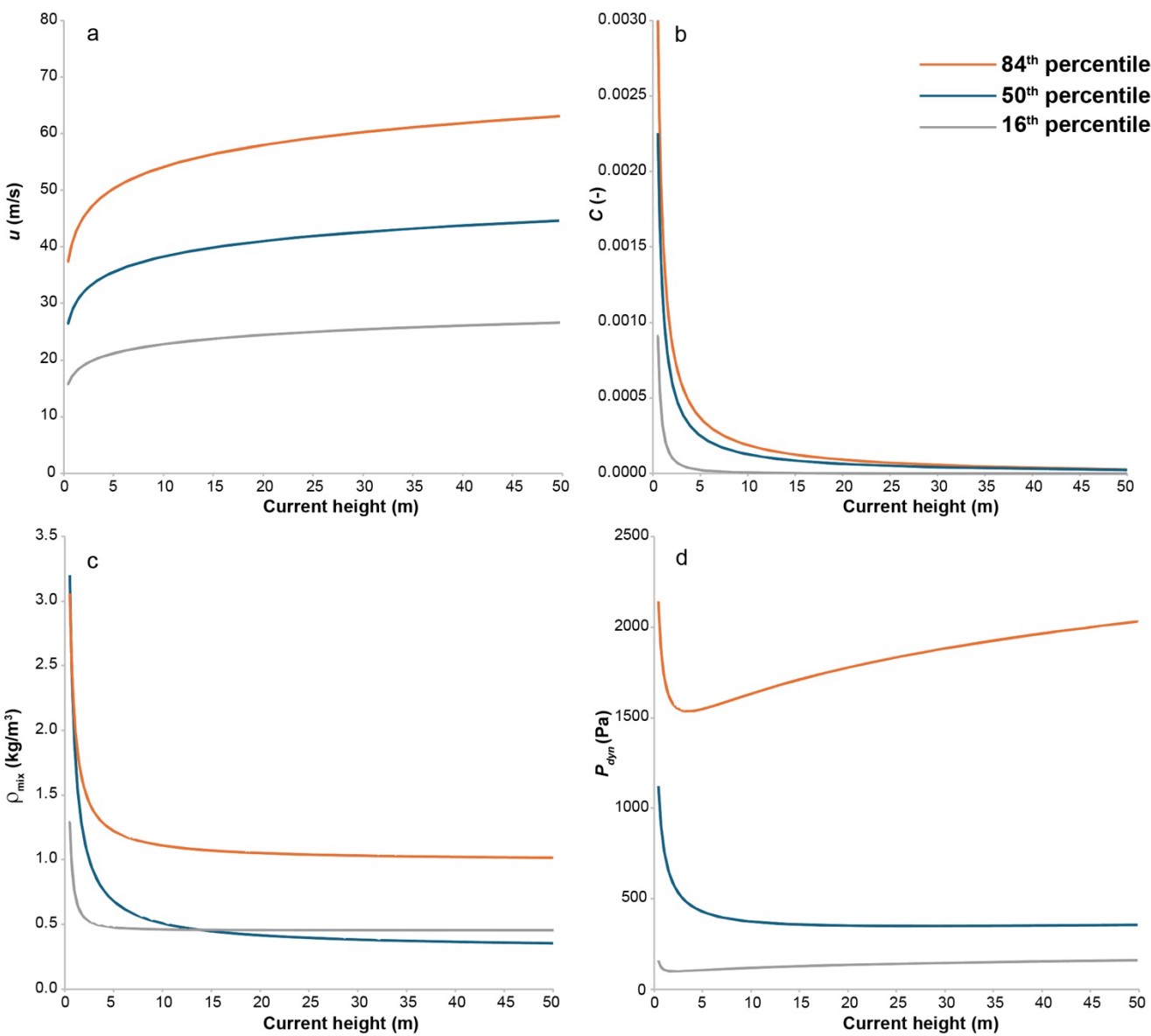

**Figure 4. Example of parameters calculation of a PDC of Pomici di Mercato eruption. The curves refer to the minimum (16th percentile), the average (50th percentile) and the maximum (84th percentile) of the probabilistic model solution. (a) Velocity profiles. (b) Particle volumetric concentration profiles. (c) Density profiles. (d) Dynamic pressure profiles.**

Velocity ($u$) logarithmically increases upward in the flow (Figure 4a), reaching values in the range of tens m/s (a list of symbols is provided in Table 1). Particle volumetric concentration ($C$) decreases with height (Figure 4b), and already in the first few meters is lower than 0.001. The density profile ($\rho_{mix}$) mimics the trend of the concentration profile (Figure 4c), and rapidly decreases down to a value lower than atmosphere (which is reached, typically, at a concentration about 0.0001), making the

upper part of the current buoyant. The dynamic pressure $P_{dyn}$ has a maximum in the first few meters (Figure 4d). Higher in the current, dynamic pressure ranges around 1 kPa. The $P_{dyn}$ value integrated over the first 10 m of the current, which we consider helpful for representing the stress acting on a typical building, is 1.7 kPa, in the 84[th] percentile curve for this example. With such a value, moderate mechanical damages are expected (Zuccaro et al., 2008; Zuccaro and Leone, 2012).

210

**Table 1. Notation**

| Symbol | Description | Units |
|---|---|---|
| $A_r$ | Aggradation rate | m s$^{-1}$ |
| $C$ | Particle volumetric concentration | - |
| $C_0$ | Reference particle concentration in the Rouse equation | - |
| $C_d$ | Drag coefficient | - |
| $C_{ga}$ | Volumetric concentration of the entrained air | - |
| $C_{ga,rel}$ | Relative volumetric concentration of the entrained air | - |
| $C_{gm}$ | Volumetric concentration of the magmatic gas | - |
| $C_{gm,rel}$ | Relative volumetric concentration of the magmatic gas | - |
| $Cp_m$ | Specific heat at constant pressure of the magmatic gas | J kg$^{-1}$ K$^{-1}$ |
| $Cp_s$ | Specific heat at constant pressure of the particles | J kg$^{-1}$ K$^{-1}$ |
| $d$ | Particle size | m |
| $d_{ent}$ | Entrained particle size | m |
| $g$ | Gravity acceleration | m s$^{-2}$ |
| $H_{dep}$ | Total deposit thickness | m |
| $H_{lam}$ | Thickness of the laminated layer | m |
| $k$ | Von Karman's constant | - |
| $k_s$ | Substrate roughness | m |
| $P_{dyn}$ | Dynamic pressure | Pa |
| $P_n$ | Particle Rouse number | - |
| $P_{n,avg}$ | Average Rouse number of the particles in the current | - |
| $P_{n,susp}$ | Average Rouse number of particles in turbulent suspension | - |
| $P_n^*$ | Normalized Rouse number of the current | - |
| $P_{ni}$ | Rouse number of the i[th] solid fraction in the deposit | - |
| $R_a$ | Specific gas constant of air | J kg$^{-1}$ K$^{-1}$ |
| $Re_*$ | Particle Reynolds number | - |
| $R_m$ | Specific gas constant of the magmatic gas | J kg$^{-1}$ K$^{-1}$ |

| $S_r$ | Sedimentation rate | kg m$^{-2}$ s$^{-1}$ |
|---|---|---|
| $T$ | Flow temperature | K |
| $t$ | Time of deposition | s |
| $T_a$ | Air temperature | K |
| $T_m$ | Temperature of the magmatic gas | K |
| $T_s$ | Particle temperature | K |
| $u$ | Velocity | m s$^{-1}$ |
| $u_*$ | Shear velocity | m s$^{-1}$ |
| $w_t$ | Particle settling velocity | m s$^{-1}$ |
| $z$ | Vertical coordinate | - |
| $z_0$ | Reference height in the Rouse equation | m |
| $z_{sf}$ | Shear flow thickness | m |
| $z_{tot}$ | Total flow thickness | m |
| $\alpha$ | Substrate slope | ° |
| $\theta$ | Shields parameter | - |
| $\mu$ | Fluid viscosity | Pa s |
| $\rho_{atm}$ | Atmospheric density | kg m$^{-3}$ |
| $\rho_g$ | Gas density | kg m$^{-3}$ |
| $\rho_{mix}$ | PDC flow bulk density | kg m$^{-3}$ |
| $\rho_s$ | Particle density | kg m$^{-3}$ |
| $\rho_{s,ent}$ | Entrained particle density | kg m$^{-3}$ |
| $\rho_{si}$ | Density of the i$^{th}$ solid fraction in the deposit | kg m$^{-3}$ |
| $\tau$ | Flow shear stress | Pa |
| $\tau_0$ | Yield strength | Pa |
| $\phi_i$ | Weight fraction of the i$^{th}$ solid fraction in the deposit | - |
| $\rho_{dep}$ | Deposit density | kg m$^{-3}$ |

### 3.1.2 Flow temperature

Flow temperature was calculated by using as input in eq. A17 (see the appendix A) the values of density, concentration, temperature and specific heat of the three components of the gas particle mixture, namely: magmatic gas, air and volcanic particles. The temperature of magmatic gas $T_m$ and of volcanic particles was set to 850 °C, which is compatible with the temperature of Vesuvius magmas (Cioni et al. 2004). Average density was set to 1700 kg m$^{-3}$ for the volcanic particles, to 0.2 kg m$^{-3}$ for volcanic gas at 850 °C, and to 1.2 kg m$^{-3}$ for air at 18 °C, respectively. The specific heats were set to 2200 J kg$^{-1}$ K$^{-}$

[1] for volcanic gas, 700 J kg$^{-1}$ K$^{-1}$ for the volcanic particles and 1005 J kg$^{-1}$ K$^{-1}$ for air. As for the particle concentration, an average value of 0.001 was set, which was obtained by integrating the concentration profile over flow height from the ground to 10 meters above the ground (see Figure 4b) by means of Eq. (A7). The relative concentrations of air and magmatic gas were obtained by the method illustrated in Appendix A and resulted as 0.941 and 0.058, respectively. A temperature about 500 °C was obtained, in the first few meters of the current, by solving Eq. (A17). The low temperature obtained in the distal areas for other deposits (sometimes lower than 200° C) is due to the very low content of solid particles and a high content of cold atmosphere air in the current, which is attributed to the air entrainment process that characterizes PDCs along runout (Dellino et al., 2019) . We consider it as a minimum value, since in our model no heat transfer between phases is considered, assuming that the time of mixing was relatively low and did not allow rapid heat exchange. It is also confirmed by experiments, as reported in a previous section.

### 3.1.3 Flow duration

Based on the assumption that the sedimentation rate of a stratified layer is almost constant during aggradation, flow duration was calculated by dividing layer thickness ($H_{dep}$) by the sedimentation rate ($S_r$) (Lajoie et al., 1998). The method is described in Appendix A, and is derived by Dellino et al. (2021). The input data (particle concentration, Rouse number and settling velocity) are all functions of the shear flow density, which was calculated in terms of a PDF with PYFLOW v2.5 (Dioguardi and Mele, 2018). As a consequence, also flow duration is expressed in terms of probabilities. The average flow duration was about 20 min, representing the case study of Pomici di Mercato eruption. The duration is quite long when compared to the couple of minutes considered as a survivable time for people engulfed in a PDC, even at low temperature (Horwell and Baxter, 2006; Baxter et al., 2017).

### 3.2 Model of the highly concentrated undercurrent that formed the massive bed

In order to constrain the general model of the basal part of PDCs that forms the massive deposits, we can start from the experimental data on granular flows of volcanic material passing over a break in slope (Sulpizio et al. 2016). The method was successfully tested against granular avalanches of 1944 eruption at Vesuvius and for some of the volcaniclastic flows that occurred on May 5-6, 1998, in the Sarno area. In particular, Sulpizio et al. (2016) provided an equation linking velocity and distance travelled beyond the break in slope, using different slope ratios:

$$\frac{v}{v_{max}} = 1 + m\mathrm{D} + n\mathrm{D}^2 + p\mathrm{D}^3 \tag{1}$$

where $v_{max}$ is the velocity at break in slope, D is the distance beyond the break in slope, and $m$, $n$ and $p$ are parameters depending by ΔH, defined as the difference in height between the source area and the front of the deposit:

$$\begin{cases} m = \dfrac{a}{\Delta H} \\ n = \dfrac{b}{\Delta H} \\ p = \dfrac{c}{\Delta H} \end{cases} \qquad (2)$$

where a, b and c are parameters depending on the slope ratio (SR), defined as the ratio between the slope downvalley and upvalley of the break in slope:

255

$$\begin{cases} -a = 4.91e^{-2.1SR} \\ b = 15.56e^{-3.51SR} \\ -c = 16.73e^{-4.88SR} \end{cases} \qquad (3)$$

In order to get the velocity at different distances beyond the break in slope we have to set $\Delta H$ and $v_{max}$. For the case under study, the elevation of impact on the ground of the collapsing pyroclastic material was set around 800-900 m (the hypothetical height of collapse from Plinian column; Wilson et al. 1980; Woods, 1995) and the elevation of deposits occurrence was set around 200-300 m, which resulted in a $\Delta H$ of 600-700 m. The velocity at the break in slope can be set around 15 m s$^{-1}$, similar to that measured for volcaniclastic flows of May 5-6, 1998 in the Sarno area (Zanchetta et al. 2004). The present-day SR around Vesuvius is close to 0.5, which has been used as input value in equation (3). Figure 5 shows the results for a $\Delta H$ of 600 m. It is worth noting that the deposit lateral extent from the break in slope, for SR=0.5, is around 800-900 m, quite in agreement with the field data (Gurioli et al. 2010).

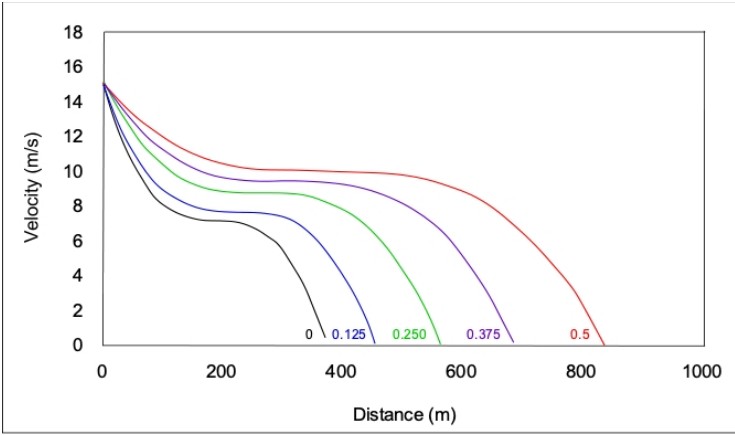

Figure 5. Velocity profiles beyond the break in slope for different SR.

270

In order to constrain the specific flow model of Pomici di Mercato eruption, we used data from the stratified layer formed by the overlying current. When cropping out on the gentle slope of the volcano flank, the stratified layer is 0.5 m thick. When it occurs along the slopes of the gullies departing from the crater rim, it is split into two parts by the intercalated massive layer of the underflow. The split consists of a 0.15 m thick basal part (Figure 3c) and a 0.35 m thick top part (Figure 3d). The intercalated massive layer is 1.6 m thick (Figure 3c). The different speeds of the two flows moving downslope justify such interpenetrating stratigraphy between the massive and the dune-bedded layer. The dilute current, being faster, overtook the slower basal flow and started forming the dune-bedded layer by aggradation at the split location. After the concentrated flow passed over, the aggradation of the dune-bedded layer continued as long as the current was fed from the source. This kind of sandwiching stratigraphy is quite common for sedimentary deposits formed by density currents, as reported for turbidites (e.g., Talling et al., 2004).

The speed of the underflow can be estimated by the ratio between the distance from the crater rim (Mount Somma) and the total time the underflow took to reach the dune-bedded layer at the 'split' location. The total time is evaluated by summing the time the overcurrent took to reach the split location plus the time that the overcurrent took to accumulate, by aggradation, the part of the stratified layer found under the massive one (0.15 m). The former time, 95 sec, was calculated by the distance, 4 km, divided by the speed of the overcurrent, ca 42 m s$^{-1}$, which was calculated by means of the turbulent boundary-layer shear flow approximation (Eq. (A6)) using PYFLOW v2.5 (see sample 13/4, in Table 2 and Mele et al. (2024)). With the software, the time of aggradation of the stratified layer found under the massive layer was calculated, and resulted in 1,140 sec. The total time that the underflow took to reach the split location was 1,235 sec, corresponding to a velocity of the massive undercurrent of 3.23 m s$^{-1}$, which is much slower than that of the overcurrent, as it is expected for a highly concentrated massive flow moving downslope.

The concentrated undercurrent stops at the base of the volcanic cone where, as a consequence of the decrease of the slope angle, it freezes in a 2 m thick massive layer (Figure 3b). Such a behaviour is typical of particulate material with a high internal yield strength that does not allow downslope flowage until a minimum shear stress is overcome, similar to a Bingham-plastic (Furbish, 1997). Such flows stop when the slope decreases, and the yield strength equals shear stress. Assuming that flow density was not much different from that of deposit, i.e., 1400 kg m$^{-3}$ (as calculated by considering a known volume of deposits and weighing it), the yield strength $\tau_0$ can be equated to the shear stress acting on the slope, which results in the minimum stress for the down-slope movement of the massive flow:

$$\tau_0 = \rho_{dep} g \sin \alpha H_{dep} \tag{4}$$

With a slope angle $\alpha$ of 1.5° and a deposit thickness $H_{dep}$ of 2 m, the yield strength is 700 Pa.

By inverting the equation of the height-averaged velocity of a Bingham-plastic

$$\overline{u(z)} = H \left( \frac{\rho_{dep} g \sin \alpha H}{3\mu} - \frac{\tau_0}{2\mu} \right) \tag{5}$$


and using the value of yield strength $\tau_0$ previously obtained, the thickness $H$ of the massive layer, and a slope angle at the split location of 6.5°, a viscosity $\mu$ of 200 Pa s results, which completes the rheological characterization of the massive underflow. Such a rheology is compatible with other massive sedimentary flows, to which massive pyroclastic flows have already been

compared in the literature (Fink et al., 1981; Major and Pierson, 1992; Palladino and Valentine, 1995; Major and Iverson, 1999; Capra et al., 2018). While such type of flows maintains mobility only inside channels and stop at the gully apron, they are still destructive at the foot of the volcano because of a dynamic pressure over of 7 kPa, and high temperature, which is due to the high particle concentration.

**4 Hazard intensity maps and expected impact**


The PDF of the impact parameters of the PDCs were reconstructed from the deposits of all eruptions that showed, in the field, a good enough exposure to both characterize the deposit structure and to sample the pyroclastic material for the laboratory analyses. Multiple deposits cropping out along the dispersal area were investigated for each eruption; specifically, we took into account 65 samples of 16 PDCs' deposits. A list of the locations of the deposits taken into analysis is provided in Table 2

and a map is displayed in Figure 6. The input and output files of all the PYFLOW simulations for each deposit is provided in Mele et al. (2024). The models used in the PYFLOW code for calculating the PDF of the impact parameters are the same as those illustrated in the previous section's example of the Pomici di Mercato eruption (Appendix A).

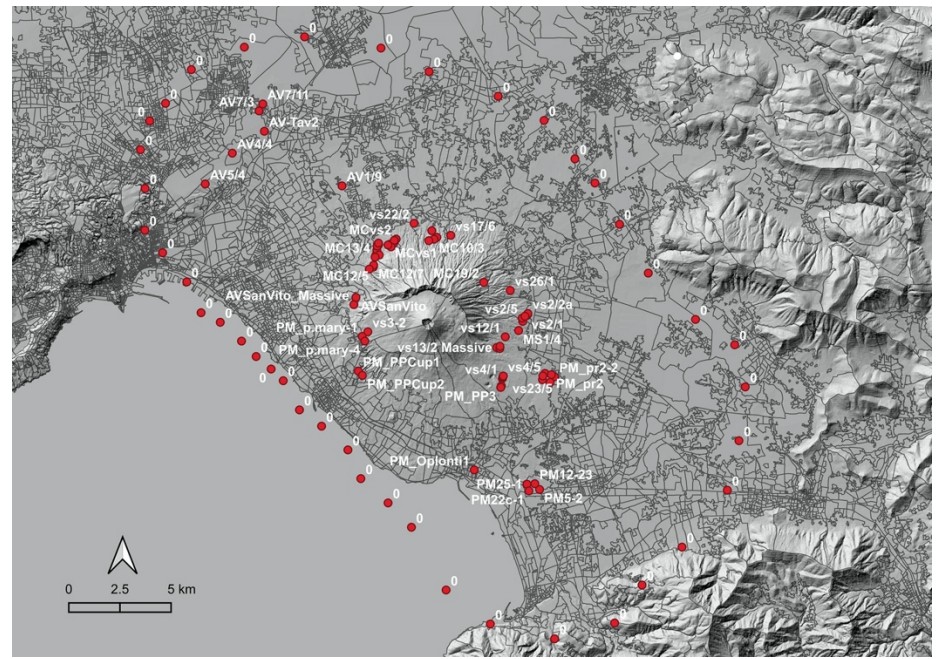

**Figure 6. Map of sample locations at Vesuvius (see Table 2 for more details). The digital elevation model (Tarquini et al., 2023),**
**territorial bases and census variables (Istat, 2011) are used as the topographic base for data set visualizations.**


**Table 2. List of the sample locations (cartographical reference system WGS 84-UTM 33N.**

| Eruption | Unit | sample | X (m) | Y (m) |
|---|---|---|---|---|
| AD 1631<br>(Rosi et al., 1993) | 1631 | MS1/4 | 456008.13 | 4518724.03 |
| Pollena (AD 472)<br>(Sulpizio et al., 2005) | S1 | vs17/6 | 452734.67 | 4523363.56 |
| | | vs19/9 | 451810.00 | 4523595.88 |
| | | vs22/2 | 450944.56 | 4523960.21 |
| | | vs2/1 | 456416.53 | 4519450.54 |
| | | vs4/1 | 455205.03 | 4516400.34 |
| | S2 | vs23/5 | 457160.09 | 4516331.52 |
| | | vs2/2a | 456466.04 | 4519547.75 |
| | | vs4/3 | 455224.57 | 4516447.74 |

| | | | | |
|---|---|---|---|---|
| | | vs23/2 | 457284.24 | 4516635.62 |
| | | vs13/2 Massive | 454938.93 | 4517904.16 |
| | | vs13/2 | 455088.76 | 4517858.92 |
| | | vs2/3 | 456213.00 | 4519171.41 |
| | | vs12/1 | 455358.48 | 4518422.88 |
| | | vs23/4 | 457179.51 | 4516514.30 |
| | Sy | vs26/1 | 455613.39 | 4520674.32 |
| | | vs4/6 | 455228.34 | 4516482.65 |
| | | vs2/5 | 456186.41 | 4519330.02 |
| | | vs13/3 | 455098.74 | 4517964.11 |
| | | vs4/5 | 455252.46 | 4516521.57 |
| | | vs2/4a | 456327.61 | 4519414.50 |
| Pompeii (AD 79) (Cioni et al., 1992) | EU3pf | PM_PP2 | 455153.96 | 4516044.29 |
| | | PM_pr2-2 | 457748.39 | 4516468.36 |
| | | PM_pr2 | 457596.00 | 4516572.87 |
| | EU4 | PM_PP3 | 455125.20 | 4515969.86 |
| | | PM_p.mary-1 | 448383.07 | 4518488.93 |
| | | PM_PR4 | 457532.66 | 4516201.35 |
| | | PMvs1_Massive | 449818.91 | 4522825.06 |
| | | PMvs1 | 449669.96 | 4522919.38 |
| | | PM_Oplonti1 | 453805.10 | 4511968.91 |
| | | PM_CPollena | 448750.75 | 4521753.94 |
| | | PM_PPCup1 | 448182.52 | 4516805.40 |
| | | PM_PPCup2 | 448379.34 | 4516597.53 |
| | | PM5-2 | 456987.12 | 4511000.96 |
| | | PM25-1 | 456374.22 | 4511259.25 |
| | | PM22c-1 | 456467.72 | 4510935.56 |
| | | PM12-23 | 456757.24 | 4511281.08 |
| | EU7 | PM_p.mary-4 | 448507.65 | 4518263.67 |
| AP2 (Cioni et al., 2008) | AP2 | vs3-2 | 448658.59 | 4518710.98 |
| Pomici di Avellino (Sulpizio et al., 2010) | EU5 a | AV2/2 | 449209.95 | 4522930.73 |
| | EU5 b | AV2/5 | 449142.01 | 4522869.39 |

| | | | |
|---|---|---|---|
| | AV2/7 | 449208.65 | 4523011.56 |
| | AV1/9 | 447451.73 | 4525799.95 |
| Cava San Vito | AVSanVito_Massive | 447987.83 | 4520035.70 |
| | AVSan Vito | 448091.94 | 4520384.36 |
| EU5 b | AV_Tav2 | 443690.19 | 4528476.65 |
| | AV4/4 | 442120.93 | 4527426.14 |
| | AV5/4 | 440790.89 | 4525942.00 |
| | AV7/3 | 443439.30 | 4529463.26 |
| EU5 c | AV7/11 | 443623.49 | 4529801.35 |
| | AV3/12_Massive | 449968.79 | 4523028.51 |
| | AV3/12 | 449978.44 | 4523138.63 |
| Pomici di Mercato (Mele et al., 2011) | MCvs1 | 449777.76 | 4522964.23 |
| | MCvs2 | 450072.39 | 4523203.00 |
| | MC13/3 | 449064.39 | 4522395.94 |
| | MC8/3 | 452056.53 | 4523272.95 |
| EU4 | MC10/3 | 451918.93 | 4523181.88 |
| | MC10/3 Massive | 451662.41 | 4523113.25 |
| | MC14/1 | 449126.76 | 4522613.68 |
| | MC14/2 | 449129.16 | 4522678.89 |
| | MC19/2 | 454328.68 | 4521073.78 |
| EU6 | MC13/4_Massive | 449242.12 | 4522437.32 |
| | MC13/4 | 449036.04 | 4522328.96 |
| | MC12/5 | 448943.77 | 4521846.17 |
| | MC12/7 | 448965.37 | 4521944.05 |

Upon processing data with the PYFLOW code, only results that were significant after a t-test on grain size, at 5% probability, were included in the final database (see PYFLOW user manual for more details). The resulting dataset consists of 65 georeferenced data points distributed around the volcano, each containing values of the four impact parameters: dynamic pressure, particle volumetric concentration, temperature and flow duration. The 84[th] percentile of the PDF, which we consider as a safety value of the intensity of PDCs, is used for constructing the hazard intensity maps.

By the analysis of results shown in Figure 7, which are arranged in order of eruption age, no temporal trend of PDC intensity (as expressed here by the dynamic pressure) emerges at Vesuvius. The variability inside an eruption (between different PDCs)

covers a broad range as it is also the variation of parameters among eruptions. Therefore, there is no reason to choose one specific eruption as representative of the hazard of PDCs in the next fifty years. Also note that if one considers the scale of eruptions as represented by the total volume of volcanic material emitted including deposits of other origin with respect to PDCs, such as Plinian fallout), which is an often used metric in volcanology, there is not any correlation between eruption size

and PDC intensity. An example is the Pollena eruption, whose PDCs are as intense as those of Pomici di Mercato or 79 AD eruptions, but have a total volume five times smaller (Sulpizio et al., 2005; 2007). All data points of all eruptions calculated at an exceedance probability of 16% (which is the complement to the 84[th] percentile of solution of the PDF) have been, therefore, used for drawing the hazard intensity maps, without any choice of a particular case as a specific scenario to be expected in the long term. Therefore, the maps represent the value of the impact parameters at an exceedance probability of 16% in the event

of a PDC-forming eruption at Vesuvius, without making assumptions on any scenario (e.g., eruption size). In the construction of the hazard intensity maps, all data points, referring the outcropping deposits of all the eruptions were considered together as to obtain isolines of the impact parameters.

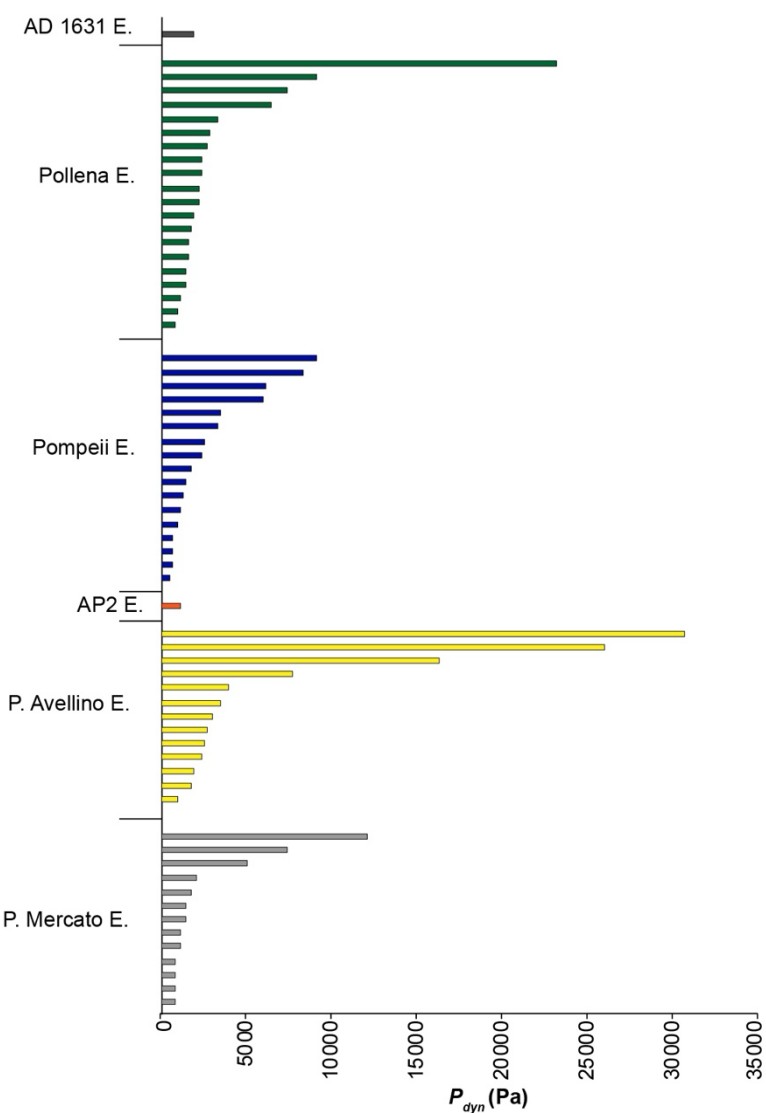

**Figure 7. Dynamic Pressure (Pa) values (in the basal 10 m of the current), calculated from the deposits of all studied eruptions. Each bar represents a location for the given eruption.**

Hazard intensity maps representing the isolines of the expected safety values of the impact parameters are shown in Figure 8. The maps were produced using the open source QGIS software. To reconstruct the maps, it was first necessary to add "zeroes" (zero values of impact parameters), representing points where PDC deposits do not crop out. While this does not completely rule out the arrival of the PDC at these locations, the potential error committed in the evaluation of the impact parameters in the areas close to the zero line is negligible since the last non-zero contour lines were set to very low values of the impact parameters. Then we applied the QGIS contour plugin in order to spatially interpolate the data (Crook and Rouberyrie, 2024).

Each map of Figure 8 represents one impact parameter. Also, data were rasterized based on a regular grid at 250 m resolution, and are provided in the Zenodo repository (Mele et al. 2024), which could be useful for vulnerability analysis.

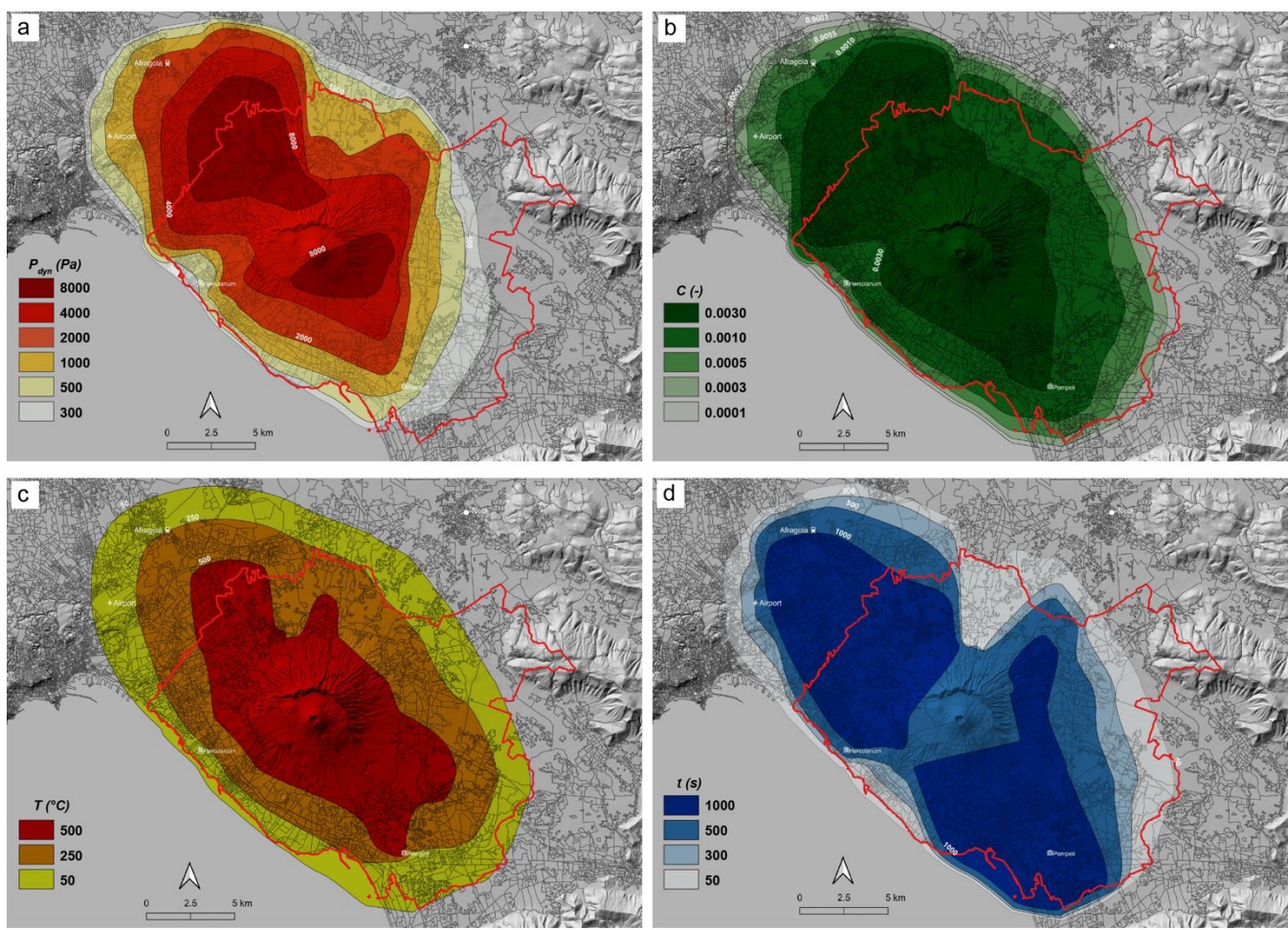

**Figure 8. Hazard intensity maps of pyroclastic density currents at Vesuvius calculated at the 84th percentile (16% exceedance probability). The red solid line represents the boundary of the red zone proposed by the Italian National Civil Protection Department**
**(2014). a) Dynamic Pressure (Pa) integrated over the basal 10 m of the current. b) Particle volumetric concentration integrated over the basal 2 m of the current. c) Flow temperature (°C) in the basal 2 meters of the current. d) Average flow duration (s). The digital elevation model (Tarquini et al., 2023), territorial bases and census variables (Istat, 2011) are used as the topographic base for data set visualizations.**

From all the maps a decrease of PDCs intensity emerges as a function of the distance from the volcano, which helps to differentiate the potential impact of PDCs over the territory compared to the undifferentiated red zone proposed by the National Civil Protection Department of Italy (Civil Protection Department, 2014; Figure 8 red solid line). Our maps tend to have similar coverage to that of the red zone, the only exception being the northwest area, where the maps extend a little further than the

red-zone limit. This is because our maps include the PDCs of the Pomici di Avellino eruption, which spread over northwest
but were not used in the drawing of the red-zone map because they were considered as representative of a too big scenario.

At the centre of the maps, around the Vesuvius cone, the massive undercurrents result in high values of impact parameters, with dynamic pressures over 8 kPa (Figure 8a) and temperature over 500°C (Figure 8c) due to the high particle concentration. These flows are totally destructive and exceed reasonable engineering measures to protect buildings and populations, aside from evacuation. This conclusion is supported by data at Herculaneum, where the massive flows of the Pompeii eruption left
a massive deposit that caused the breaking of thick Roman walls and charred wood components of buildings (Giordano et al. 2018). Results of the calculations for the massive undercurrents in the locations, where it has been possible to apply the model of section 3.2, are provided in the Zenodo repository (Mele et al. 2024). Moving away from the cone, the isolines of the impact parameters refer to the overlying dilute currents, since the massive underflow stops at the base of the volcano. Values of the impact parameters vary significantly moving away from Vesuvius, implying a different impact in the various zones around the
volcano. A constraint to the flow mobility exists toward northeast, which is represented by the remnants of Monte Somma, while toward southwest there is the sea, which is not considered in the red zone.

Focusing on the impacts on buildings, dynamic pressure (Figure 8a) shows values exceeding 8 kPa in the more proximal zones, both toward northwest and southwest, while they reduce to less than 1 kPa at the margin of the red zone. This is due to the
decrease of both speed and concentration. Engineering investigations (Spence et al., 2004; Zuccaro et al., 2008; Zuccaro and Leone, 2012) show that values higher than 5 kPa can significantly damage buildings, while pressure under 1 kPa has minimal to no consequence on structures or infrastructures. Different impacts can be indeed expected on buildings as one moves away from the volcano, and while in proximal areas severe damages are expected, at distal locations such as Pompeii, 10 km from the volcano, the mechanical effects of the dilute currents strongly decrease down to a value lower than 1 kPa. No damage to
walls should be expected with such a flow strength (Spence et al., 2004; Zuccaro et al., 2008; Zuccaro and Leone, 2012). This is consistent with the fact that at Pompeii, during the 79 AD eruption, the walls of Roman buildings do not show evidence of damage related to the passage of the PDC (Luongo et al., 2003; Gurioli et al., 2007). While this is not the proper place to discuss in detail the engineering actions that can be used for protecting existing buildings or to propose guidelines for new constructions against the impact of PDCs, our map of Figure 8a suggests that already a few km away from the volcano, but
still well inside the red zone, actions for protecting openings and walls (which are the weaker elements of buildings) against a dynamic pressure of a few kPa could be viable.
Concerning the effects of PDCs on the population caught unprotected, the combination of data from maps of particle volumetric concentration, temperature and flow duration of Figure 8 b, c, and d, respectively, allows to assess that even in distal zones, where the mechanical effect of dynamic pressure drastically decays, the effect of hot fine-ash needs to be considered as a
primary impact over the population. The ideal protective measure, even at these locations, is evacuation. In fact, it is emerging that even in areas far from a volcano, where particle concentration, temperature and dynamic pressure strongly decrease, people engulfed in the flow have "high probability of receiving fatal skin burns and inhalation injury of the upper and lower respiratory

tract, unless the duration is very brief" (Baxter et al., 2017). The presence of fine-ash particles suspended in air for a long time, even in very small amounts, can be very harmful to human health, and represents one major cause of injury (Horwell and Baxter, 2006). Our maps show that temperature decays from 500°C in the zone characterized by the massive undercurrents around the cone, which is justified by the high particle volumetric concentration, down to values lower than 200°C at the margin of the red zone (Figure 8b). This decrease is due to the large volume of cold air entrained in the current during runout. The low temperature of the PDCs of the 79 AD eruption calculated at Pompeii (about 115°C) is due to the much higher content of cold atmosphere air in the current, with respect to the hot magmatic gas. Exposure to pure hot air at 200–250 °C can be survived for 2–5 minutes (Buettner, 1950), but the presence of inhalable hot fine ash drastically reduces survival times (Baxter et al., 2017). As expected, our map of particle concentration (Figure 8c) shows an abrupt decay passing from the area around the cone, which is characterized by the massive undercurrent, to values much lower than 0.001, typical of the dilute overlying current in distal reaches. Even a volumetric concentration of ash in suspension this low can be unbreathable and is one of the main causes of mortality of PDCs. This is consistent with the observation of historical eruptions, where the flow lasted for several minutes to hours (Lube et al., 2007). At that moment, the territory surrounding the volcano was engulfed by thick, expanded, fast and hazardous currents, loaded with unbreathable hot ash (Horwell and Baxter, 2006).

The exposure time becomes indeed a major factor in determining the impact of PDCs on population, since it quantifies the residence time of hot volcanic ash that can be inhaled by people potentially exposed to the currents (Horwell and Baxter, 2006). Our map of Figure 8d, shows that flow duration ranges always exceed several minutes. These values refer only to the overlying dilute currents, since the massive undercurrents that freeze at the foot of the cone are much shorter lived. In the case of Pompei eruption of 79 AD a value of 17 min had been calculated, which combined with the concentration of ash particles (about 0.001), was a long enough time to cause death by asphyxia at Pompeii. There are reports of recent eruptions showing that in the marginal reaches of the current, where the flow duration was only a few minutes, people were able to survive (Baxter et al., 2017). In other cases, longer flow durations did not permit survival and death was caused by fine-ash inhalation (Baxter et al., 2017; Nakada, 2000). We agree with Baxter et al. (2017) that the emergency planning for explosive eruptions should concentrate on the distal parts of PDCs where survival could be feasible, and where the primary risk to life is asphyxiation from ash inhalation, rather than thermal or mechanical injury. It is important to take note of such information when projecting for emergency plans and risk-reduction measures.

## 5 Conclusion

Among volcanic phenomena, PDCs are the major cause of destruction and injuries in urbanized areas. Since it is impossible to predict the exact dispersal area or the magnitude of an eruption, a probabilistic approach that accounts for the variability of the intensity of pyroclastic density currents of past eruptions is a better choice to inform structural mitigation and, in case of an impending eruption, sustainable evacuation plans.

Here, we combined field data from deposits of previous eruptions with a physical model to develop probability density functions for impact parameters around Vesuvius. No specific eruption was chosen as representative of the hazard of PDCs, because there is no correlation between eruption size and impact parameters.

By considering the 84[th] percentile of the distribution as a safety value, hazard intensity maps have been drawn that show the distribution over the territory of pyroclastic density currents intensity in the long term. Our choice of the 84[th] percentile as a safety value could appear as a particularly severe one, but it is often used in Geophysics and in Engineering (Bradley, 2011; Fang et al., 2020). These maps differ from the red-zone map of the Italian Civil Protection Department in two main aspects:

1.  Our maps include the distribution of the PDCs' intensity (as represented by impact parameters). In contrast, the red-zone map is undifferentiated since it was constructed simply by delineating the outer margin of deposit dispersal, but not the PDCs intensities. The red-zone map is used to delimit the area to be evacuated, and not to project for possible mitigation actions.

2.  Our hazard zones extend toward northwest compared to the red-zone map, because in our case all eruptions of Vesuvius are included, while in the construction of the red-zone map the Avellino and Pompeii eruptions were not considered, because it was assumed that eruptions that big should not occur. PDC intensity is not proportional to the volume of an entire eruption, since one single PDC is a small fraction of the total volume. An example is that the bigger eruption of Pompeii had weaker currents than the smaller eruption of Pollena.

Since our maps extend a little bit more toward northwest with respect to the red-zone map of the Department of Civil Protection of Italy, it would be useful considering an extension over that direction of the evacuation zone around Vesuvius.

In this study, an integrated model resolving the impact parameters of both the underlying massive part and of the overlying dilute part of the current was used, allowing to differentiate their respective impact. The Bingham plastic rheology used to approximate the massive underflow is similar to that proposed for other massive flows that occur both on volcanoes and on sedimentary terrains. To our knowledge this is the first time that such an integrated approach, resolving the complexity of both the concentrated and the dilute part of the flow, is used for constructing hazard intensity maps, and deserves to be taken into consideration also on other volcanoes that show a complex stratigraphy of PDCs. Such complex stratigraphy, at Vesuvius, implies that during an explosive eruption multiple currents occur, making it reasonable to assume that PDCs can continue for hours or days or more, and that their multiple, cascading effect, need to be considered when projecting for mitigation actions.

The maps of impact parameters make it possible to back calculate the initial and boundary conditions of PDCs at the crater and to simulate, by 3D computational fluid dynamics, the propagation of currents over the actual morphology, including the urbanized area around the Vesuvius, which is the next step of the present research.

The precision of parameters used in the PYFLOW code needs to be tested against alternatives to assess the modelling approach's epistemic uncertainty. An extension of this work will be dedicated to such a subject, in order to assess the multi-model variability of results. We think that the method used here to prospect probabilistically the hazard in the long term, and to take as a safety value the 84[th] percentile of PDF, covers an ample range of the uncertainty of results.

## Appendix A. The flow model and the numerical code

The reconstruction of the impact parameters of PDCs is based on a flow model that starts with the assumption that the current is steady, velocity and density stratified (Valentine, 1987; Dellino et al., 2008; Brown and Branney, 2013) and flowing on gentle slopes. The model is implemented in the Fortran numerical code PYFLOW v2.5 (Dioguardi and Dellino, 2014; Dioguardi and Mele, 2018).

In the stratified multiphase gas-particle current, the basal part is a shear flow that moves attached to the ground and has a density higher than atmosphere. The upper part is buoyant, because particle concentration decreases with height down to a value that, combined with the effect of gas temperature, makes the mixture density lower than the surrounding atmosphere.

The inputs needed, in our model, for the calculation of the impact parameters are reported in the input files of the Zenodo repository (Mele et al. 2024). Some of the input data are obtained directly in the field, such as deposit and layer thickness. Deposit density is obtained by weighing a known volume of deposit. Other data come from laboratory analyses on samples extracted from the deposit. In the laboratory, first, the grain-size distribution is determined, then from each size class a sample of particles per each component (crystal, glass, lithics) is extracted, and density data are obtained on such particle samples by means of pycnometers (Mele et al., 2015). Particle shape parameters, which are needed for the calculation of settling velocity, are obtained by image analysis methods (Mele et al., 2011).

In a dilute PDC, particles are mainly transported by turbulent suspension and sedimentation is controlled by a balance between flow shear velocity $u_*$, which is controlled by fluid turbulence and favours suspension, and particle settling velocity $w_t$:

$$w_t = \sqrt{\frac{4gd(\rho_s - \rho_{mix})}{3C_d\rho_{mix}}} \quad \text{(A1)}$$

which favours sedimentation, where $g$ is gravity acceleration, $d$ is particle size, $\rho_s$ is the particle density, $\rho_{mix}$ is the bulk flow density and $C_d$ is drag coefficient. The median of the grain-size distribution was used for particle size. PYFLOW allows selecting among multiple shape-dependent drag laws; in this work, the drag law of Dioguardi et al. (2018) was used. The capacity of a current to transport particles in suspension is quantified by the Rouse number (Rouse, 1939) $P_n = \frac{w_t}{ku_*}$, where $k$ is the Von Karman constant (0.4). At the limit of transportation by turbulent suspension when $P_n = 2.5$, from its defitinion, since k=0.4, it follows that:

$$w_t = u_* \quad \text{(A2)}$$

This is the suspension-sedimentation criterion (Middleton and Southard, 1984), which means that particles stay suspended until their settling velocity is less than the flow shear velocity. In other terms, particles in the deposit that are settled from suspension (the laminae-forming bed load) give an indication of the current shear velocity, once their terminal velocity is defined. Upon combining (A1) and (A2), it follows that:

$$u_*^2 = \frac{4gd(\rho_s - \rho_{mix})}{3C_d\rho_{mix}} \quad \text{(A3)}$$

which leads to the shear stress at the base of the current:

$$\tau = \rho_{mix}u_*^2 \quad \text{(A4)}$$

There can be also particles that are never transported in suspension but can be moved over the substrate by the overlying current's shear stress (e.g., particles for which $P_n > 2.5$ or that are already on the ground before the passage of the dilute PDC). The latter phenomenon can be described by the Shield or entrainment criterion (Miller et al., 1977), which compares the dilute PDC shear stress to the buoyancy force of the coarse particle in the flow:

$$\theta = \frac{\rho_{mix} u_*^2}{(\rho_{s,ent} - \rho_{mix}) g d_{ent}} \tag{A5}$$

where $\rho_{s,ent}$ and $d_{ent}$ are the density and diameter of the entrained particle, respectively; $\theta$ is a parameter which is equal to 0.015 for a particle Reynolds number $Re_* = \frac{\rho_{mix} u_* d_{ent}}{\mu}$ (where $\mu$ is the fluid viscosity) larger than 1000 (Miller et al., 1977), a condition that holds for most dilute PDCs (Dellino et al. 2008).

Both methods are implemented in PYFLOW v2.5 and can be alternatively activated depending on the PDC's deposit's architecture. When the typical complete stratigraphic sequence attributed to a dilute DPDC is observed (e.g., Figure 2c), that is:

1. a coarse layer of lapilli and bombs moved by shear at the base of the current;
2. a laminated layer of ash formed by particles settled from turbulent suspension;

it is possible to apply both the Shield and the suspension-sedimentation criteria for calculating the flow parameters. However, the layer of entrained coarse lapilli or bombs, which is typical of proximal locations around the eruptive vent, is often missing in distal outcrops, thus preventing to use the Shield criterion far away from the volcanic vent. In that case, an alternative method based on the hydraulic equivalence of particles can be used.

In both cases the parameters needed to calculate the vertical profiles of velocity, particle concentration (hence flow density), flow temperature and dynamic pressure are obtained. Specifically, the velocity profile $u(z)$ follows the equation of a turbulent boundary layer shear flow moving over a rough surface (Furbish, 1997):

$$\frac{u(z)}{u_*} = \frac{1}{k} \ln \frac{z}{k_s} + 8.5 \tag{A6}$$

where $k_s$ is the roughness parameter of the substrate. The concentration profiles is taken from Rouse (1939):

$$C(z) = C_0 \left( \frac{z_0}{z_{tot} - z_0} \frac{z_{tot} - z}{z} \right)^{P_n} \tag{A7}$$

in which $z_{tot}$ is the total flow thickness, $z_0$ is the height at which the particle concentration is known ($C_0$). From (A7), the flow bulk density profile can be defined as:

$$\rho_{mix}(z) = \left(1 - C(z)\right)\rho_g + C(z)\rho_s \tag{A8}$$

PYFLOW first estimates the shear flow height $z_{sf}$ by solving the system of equations composed of eq. (A4) and:

$$\tau = (\rho_{mix} - \rho_{atm}) g \sin \alpha \, z_{sf} \tag{A9}$$

where $\rho_{atm}$ is the atmospheric density and $\alpha$ is the slope of the ground, measured in the field, on which the dilute PDC was flowing.

The shear current is composed of gas and a mixture of particles, in which those with $P_n = 2.5$ are at settling condition. Finer particles are held in suspension by turbulent motion and contribute to the concentration profile $C(z)$, but their average Rouse

number $P_{n,susp}$, which is lower than 2.5, is unknown. In addition, the thickness of the PDC $z_{tot}$ and the flow gas density $\rho_g$ are unknown. In order to get these three unknowns, PYFLOW solves for the following system of three equations:

$$\rho_{atm} = \rho_g + \left( (\rho_s - \rho_g)C_0 \left( \frac{z_0}{z_{tot}-z_0} \frac{z_{tot}-z_{sf}}{z_{sf}} \right)^{P_{n,susp}} \right) \tag{A10}$$

$$\rho_{mix} = \frac{1}{z_{sf}-z_0} \int_{z_0}^{z_{sf}} \left[ \rho_g + \left( (\rho_s - \rho_g)C_0 \left( \frac{z_0}{z_{tot}-z_0} \frac{z_{tot}-z}{z} \right)^{P_{n,susp}} \right) \right] dz \tag{A11}$$

545 $$z_{tot} = \frac{H_{lam}}{C} = \frac{z_{lam}}{\frac{\rho_{mix}-\rho_g}{\rho_s-\rho_g}} \tag{A12}$$

The first equation (A10) states that the atmospheric density is reached at the top of the shear flow $z_{sf}$; the second one (A11) defines the average flow density calculated between $z_0$ and $z_{sf}$; the third equation (A12) defines the total flow thickness as the ratio between the thickness of the laminated layer $H_{lam}$ in the deposit and the average concentration in the flow C (Lajoie et al., 1998), which is defined as:

550 $$C = \frac{\rho_{mix}-\rho_g}{\rho_s-\rho_g} \tag{A13}$$

This is just an approximate value used to initialize the software and does not influence much the solutions down to the first tens of meters of the current where the maximum intensity of impact parameters is found. In this work $C_0$ is set to the maximum packing for pyroclastic particles (0.7) (Dellino et al., 2008), hence $z_0$ is taken as the minimal sedimenting thickness.

Subsequently, PYFLOW uses $\rho_g$ to calculate the flow temperature profile $T(z)$, assuming the flow is composed by the solid

555 particles, the magmatic gas and entrained air, if the user provides in input: the temperature of the magmatic gas $T_m$, the air temperature $T_a$ (set by default to 293 K if not provided), the temperature of the solid particles $T_s$, the specific gas constant of the magmatic gas $R_m$ and air $R_a$ (set by default to 287 J kg$^{-1}$ K$^{-1}$), the specific heat at constant pressure of the magmatic gas $Cp_m$ and of the solid particles $Cp_s$ and the average density of the solid particles $\rho_s$. First, the density of the magmatic gas and entrained air are obtained by solving for the equation of state:

560 $$\rho_m = \frac{p_a}{R_a T_a} \tag{A14a}$$

$$\rho_a = \frac{p_a}{R_m T_m} \tag{A14b}$$

hence with the assumption that the gas phases are at constant atmospheric pressure (set to 101325 Pa if not specified in input by the user). From these densities and the flow gas density $\rho_g$, one can calculate the relative volumetric concentration of the magmatic gas $C_{gm,rel}$ and entrained air $C_{gm,rel}$:

565 $$C_{gm,rel} = \frac{\rho_g - \rho_a}{\rho_m - \rho_a} \tag{A15a}$$

$$C_{ga,rel} = 1 - C_{gm} \tag{A15b}$$

These concentrations are still not the real magmatic gas $C_{g,m}$ and entrained air $C_{g,a}$ volumetric concentrations in the multiphase flow that includes the solid particle concentration calculated C via Eq. (A13), hence they need to be rescaled so that the sum of their rescaled values equals $1-C$:

$$C_{g,m} = C_{gm,rel}(1 - C) \tag{A16a}$$

$$C_{g,a} = C_{ga,rel}(1 - C) \tag{A16b}$$

Finally, the flow temperature can be calculated using the following equation, which neglects the heat transfer between particles and the fluid in a mixture model-like approach (see, e.g., Cerminara et al. 2016) and attributes the change in temperature mainly to the air entrainment and flow dilution:

$$T(z) = \frac{\rho_m C_{g,m}(z)T_m Cp_m + \rho_a C_{g,a}(z)T_a Cp_a + \rho_s C(z)T_s Cp_s}{\rho_m C_{g,m}(z)Cp_m + \rho_a C_{g,a}(z)Cp_a + \rho_s C(z)Cp_s} \tag{A17}$$

By combining the velocity (A6) and density (A8) profiles, the dynamic pressure profile is finally obtained:

$$P_{dyn}(z) = \frac{1}{2}\rho_{mix}(z)u(z)^2 \tag{A18}$$

Concerning flow duration, in a PDC, sedimentation occurs at a rate $S_r$ that represents the mass of particles settling over a unit area in the unit time. Deposit thickness is the result of the aggradation of particles during the time-integrated passage of the current. The aggradation rate $A_r$, which is the rate at which deposit thickness grows, is equal to the sedimentation rate divided by deposit density $\rho_{dep}$. The total time of aggradation, $t$, which is a proxy of flow duration, is equal to deposit thickness $H_{dep}$ divided by $A_r$:

$$t = \frac{H_{dep}}{A_r} \tag{A19}$$

Deposit density and thickness are measured in the field, consequently the only missing quantity for the calculation of flow duration is the sedimentation rate.

Dellino et al. (2019), recently proposed a model for the calculation of the sedimentation rate:

$$S_r = \left( \sum_i^n \rho_{s_i} w_{t_i} \left( \frac{\frac{\phi_i/\rho_{s_i}}{\sum_{i=1}^n \phi_i/\rho_{s_i}}*C_{tot}}{\left( \left( 10.065*\frac{P_{ni}}{P_n^*} \right)+0.1579 \right)} * 0.7 + \frac{\frac{\phi_{i+1}/\rho_{s_i+1}}{\sum_{i=1}^n \phi_{i+1}/\rho_{s_{i+1}}}*C_{tot}}{\left( \left( 10.065*\frac{P_{ni}}{P_n^*} \right)+0.1579 \right)} * 0.3 \right) \right) - 0.01 \tag{A20}$$

with the subscript $i$ referring to the $i^{th}$ particle-size class, $n$ being the number of size classes of the grain-size distribution of the sediment, $\phi_i$, $\rho_{si}$ and $P_{ni}$ being the weight fraction, the density and the Rouse number of the $i^{th}$ grain-size fraction, respectively. $P_n^* = \frac{P_{n,avg}}{P_{n.susp}}$ is the normalized Rouse number of the current, i.e., the ratio between the average Rouse number of the solid material in the current and the Rouse number at maximum suspension capacity. The model considers the contribution of each size class of particles to the sedimentation, and not the average grain size, because the solid load constituting a suspension current, especially in the case of PDCs, is made up of a mixture of different components (lithics, glassy fragments and crystals) with different size, density and shape, thus different terminal velocity. The average Rouse number of the solid material in the current is calculated as the average of the particulate mixture:

$$P_{n_{avg}} = \sum_{i=1}^n P_{ni}\, C_i/C \tag{A21}$$

When $P_n^* > 1$, a current has a particle volumetric concentration in excess of its maximum capacity, e.g. it is over-saturated of particles, which favours sedimentation. When it is lower than 1, a current has a particle volumetric concentration lower than its maximum capacity, e.g. it is under-saturated, and could potentially include additional sediment in suspension by erosion from the substrate. For more details see Dellino et al. (2019).

Finally, PYFLOW calculates probability density functions of all the parameters presented above starting from a Gaussian distribution. From these functions, it is possible to obtain the values of the impact parameters at the desired exceedance probability.

### Code availability

PYFLOW v2.5 is available at https://github.com/FabioDioguardi/PYFLOW/releases/tag/v_2.5.

### Data availability

All supporting data, which include input and output files of the DPDCs simulations carried out with PYFLOW, calculations of the massive PDCs' impact parameters and rasterized impact parameters map data are available at https://doi.org/10.5281/zenodo.13378963.

### Author contribution

PD developed the methodology and the models and contributed to data analysis and text editing. FD developed PYFLOW v2.5, contributed to the simulations, data analysis and text editing. DM run the simulations with PYFLOW, conducted data analysis and produced the graphical outputs and contributed to text editing. RS contributed to data analysis and text editing.

### Competing interests

The authors declare that they have no conflict of interest.

### Acknowledgements

The associate editor (Giovanni Macedonio), Greg Valentine and an anonymous reviewer greatly helped in improving the manuscript. This study was carried out within the RETURN Extended Partnership and received funding from the European Union Next-GenerationEU (National Recovery and Resilience Plan – NRRP, Mission 4, Component 2, Investment 1.3 – D.D. 1243 2/8/2022, PE0000005).

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
