# Peer review of "Long-term hazard of pyroclastic density currents at Vesuvius (Southern Italy) with maps of impact parameters"

_EGUsphere, 2024_

## Referee Comment (RC1)

Review of: **Long-term hazard of pyroclastic density currents at Vesuvius (Southern Italy) with maps of impact parameter**

This work considers four impact parameters (dynamic pressure, particle volumetric concentration, temperature, and flow duration) of PDCs at Vesuvius using the deposits of six prior (well-preserved) events. Overall, the manuscript is structured well, with sufficient mathematical detail in the appendix for a reader to follow the process.

My main concern is that the probabilistic components are not being handled correctly – specific examples include Figure 3b and c (where percentile lines are either in the wrong order or cross each other), and the final hazard maps (Figure 7) that appear to be a mishmash of some probabilities, some summation, and no consideration of conditional probabilities (e.g., the probability of a PDC given an eruption), or the potential for future events exceeding the parameters of the six PDCs used. However, I think these issues can be resolved without substantial effort.

**Main comments:**

**1 – The limitations of only 6 PDC deposits:** This may be a necessary limitation of the study, but its limits are not discussed, the deposits used were selected due to their preservation state and continuity. Presumably larger deposits are more likely to be preserved – is deposit preservation effected by location? Whether the PDC occurred before heavy rainfall? Lahars? Ash deposits occurring simultaneously? Does current topography reflect that of the region during each PDC deposition event? Etc. etc.

**2 – Claim that "there is not any correlation between eruption size and PDC intensity"** (Line 298): This is a bold statement provided without any corroborating references. Additionally, do the authors also think that there is no correlation between eruption size and PDC occurrence?

**3 – "Hazard maps":** Not convinced these are hazard maps under any of the current definitions, Figure 7 is useful, but is not a hazard map (see figure comments below).

**4 – Text:** There are quite a few places where the English/grammar needs to be corrected. I have tried to identify most of these, but I cannot guarantee I got all of these.

**Line-by-line:**

Line 9: basing → based

Line 21: in the impending of an eruption → in the likely event of an eruption? Under conditions of an impending eruption?

Line 35: affects → affect

Line 38: unvaluable → invaluable (??)

Line 40: not sure if it's "the hazard of a volcano", it's the PDC hazard during an eruption from a volcano

Lines 43-44: Sentence needs rewriting, and 700,000 not 700.000

Line 51: Suggest removing "This is the way the paper is organized"

Line 56: Pompei → Pompei's

Line 62: "outcropping continuity" probably needs a bit more explanation – how much continuity does a PDC require for its inclusion in your database?

Lines 69-71: I disagree with this statement, using only the PDCs with well preserved deposits almost definitely biases you towards the larger events and in no way are "all the suitable PDC-forming eruptions...considered"

Figure 1: A map of the locations of these deposits would be greatly beneficial, and maybe put the letters on Fig 1c in white? Hard to see in black.

Line 98: extra space between to and a

Lines 108-110: Sentence needs rewriting, and through not trough, and maybe by not from?

Line 130: help constraining → helped constrain

Line 147: ref needs fixing

Line 150 and Line 158: statistic → statistical

Figure 3: Apart from the issues with the percentile lines, I found this figure relatively hard to follow in the text – these are for a single point along the flow? Where the flow depth is 50 m? Maybe a bonus figure before this showing where the cross section is taken might help?

Table 1: Most of these are actually mainly referred to in the Appendix, I wonder if a subsection would be more beneficial here? Or move the whole table to the appendix where the bulk of the maths is anyhow?

Line 175: Is A13 the correct equation for flow temperature??

Line 183: by integration here – is this integrating over the 0 to 100%iles at 10m? Even then, 0.001 might be hard to get to?

Line 184: extra space between of and the

Line 193: remove "value of"

Line 194: What is section 14 of figure 2??

Line 201: flows occurred → flows that occurred

Line 222: DH → delta H

Lines 257 and 260: There has already been and equation (1) and (2)

Figure 5: This is quite hard to read, maybe remove the infrastructure data here or something?

Table 2: Suggest this goes to Appendix

Lines 298: As before – needs references / extra support for this statement

Line 300: "not weaker" is a bit vague – what parameters are you talking about here?

Lines 309-310: Does this addition of zeros force the contours to not enter these areas? Where were the zeros added? Is this saying that future PDCs can never reach locations they have not been to before?

Figure 7: This is a great way of showing the difference in impact parameters across a map but I cannot make the leap to "hazard map". What is the statement you would make associated with a point, e.g., at Afragola airport – what is the hazard there? Is it not "Given an eruption in the next 50 years the exceedance probability….", it is not "In the next eruption we would expect…..", what is the hazard-based statement you can get from these?

Line 322: differentiating → differentiate.

Line 380: "consider the PDCs of all eruptions" → nope, you've considered (and summed essentially) the 6 largest (assumed due to best-preserved) PDCs from previous eruptions.

Line 379: exceeds → exceed

Lines 391-392: Highly speculative here, suggest removal.

Lines 394-397: I don't think you did a "probabilistic approach that accounts for the variability of the intensity…"

Lines 408-410: Same big claim again – requires references.

Line 434: "also the higher end of epistemic uncertainty." → no idea what this means here

Line 478: extra ")" needs removing

---

## Referee Comment (RC2)

[referee-annotated manuscript omitted]

---

## Author Response (AR1)

Dear Editor,

I'm sending the revised version of the paper "Long-term hazard of pyroclastic density currents at Vesuvius (Southern Italy) with maps of impact parameters" by Dellino et al, submitted to NHESS. All the suggestions and comments of the two reviewers were taken into consideration in the revision, and the main aspects are discussed in the following point-by-point response to the two reviewers. In italics reviewer comments in plain our response.

The submission consists of 3 files: 1) this rebuttal; 2) Word file with track changes; 3) final revised manuscript.

Regards

Piero Dellino

**Point-by-point response**

In the following we list the reviewers comments in italics and our responses. When lines in the manuscript are reported, these refer to the track-changes file.

**Reviewer 1**

*My main concern is that the probabilistic components are not being handled correctly – specific examples include Figure 3b and c (where percentile lines are either in the wrong order or cross each other), and the final hazard maps (Figure 7) that appear to be a mishmash of some probabilities, some summation, and no consideration of conditional probabilities (e.g., the probability of a PDC given an eruption), or the potential for future events exceeding the parameters of the six PDCs used. However, I think these issues can be resolved without substantial effort.*

About figure 3. Indeed, we made a mistake in figure 3b, we have now corrected it. Concerning figure 3c, in some cases the lines of the flow bulk density profiles may cross depending on the shape of the Rouse profile, which is determined by the Rouse number, with the latter being different for each solution.

We agree with both the reviewers that Figure 7 may not be seen as hazard maps as defined by the reviewers. They may be described as hazard intensity maps, in which the impact parameters of all the identifiable past PDCs are taken into account. In our work, we did not want to link the hazard to a specific a-priori scenario (which we believe being a strong assumption too); instead, we calculated the probability density function of the impact parameters of all the PDCs deposits that can be observed in the field, which, in our view, represent the only measurable evidence that can be used to draw conclusion on the hazard.

*Main comments:*

*1 – The limitations of only 6 PDC deposits: This may be a necessary limitation of the study, but its limits are not discussed, the deposits used were selected due to their preservation state and continuity. Presumably larger deposits are more likely to be preserved – is deposit preservation effected by location? Whether the PDC occurred before heavy rainfall? Lahars? Ash deposits occurring simultaneously? Does current topography reflect that of the region during each PDC deposition event? Etc. etc.*

Actually we used 16 PDCs deposits from 6 eruptions. This could be inferred by Table 2 and Figure 6. However, we provided further explanation in lines 217-218

. Also, according to reviewer two comment

*2 – Claim that "there is not any correlation between eruption size and PDC intensity" (Line 298): This is a bold statement provided without any corroborating references. Additionally, do the authors also think that there is no correlation between eruption size and PDC occurrence?*

Also, at 218-220 clarifying the point about eruption size and probability of occurrence

*3 – "Hazard maps": Not convinced these are hazard maps under any of the current definitions, Figure 7 is useful, but is not a hazard map (see figure comments below).*

See our reply to this point above.

*4 – Text: There are quite a few places where the English/grammar needs to be corrected. I have tried to identify most of these, but I cannot guarantee I got all of these.*

We have put effort in improving the English grammar thanks to the suggestions of both reviewers.

*Line-by-line:*

*Line 9: basing → based*

Modified

*Line 21: in the impending of an eruption → in the likely event of an eruption? Under conditions of an impending eruption?*

Modified

*Line 35: affects → affect*

Modified

*Line 38: unvaluable → invaluable (??)*

Modified

*Line 40: not sure if it's "the hazard of a volcano", it's the PDC hazard during an eruption from a volcano*

Modified

*Lines 43-44: Sentence needs rewriting, and 700,000 not 700.000*

Modified

*Line 51: Suggest removing "This is the way the paper is organized"*

Modified

*Line 56: Pompei → Pompei's*

Modified

*Line 62: "outcropping continuity" probably needs a bit more explanation – how much continuity does a PDC require for its inclusion in your database?*

We rephrased the sentence as to clarify that to interpolate data points some continuity of pexposure is needed.

*Lines 69-71: I disagree with this statement, using only the PDCs with well preserved deposits almost definitely biases you towards the larger events and in no way are "all the suitable PDC-forming eruptions…considered"*

We agree with the reviewer that we could be more explicit here in terms of the limitations resulting from our approach. We have now included a statement about this in lines 72-75.

*Figure 1: A map of the locations of these deposits would be greatly beneficial, and maybe put the letters on Fig 1c in white? Hard to see in black.*

Agree, now on fig.6

*Line 98: extra space between to and a*

Modified

*Lines 108-110: Sentence needs rewriting, and through not trough, and maybe by not from?*

Modified

*Line 130: help constraining → helped constrain*

Modified

*Line 147: ref needs fixing*

Modified

*Line 150 and Line 158: statistic → statistical*

Modified

*Figure 3: Apart from the issues with the percentile lines, I found this figure relatively hard to follow in the text – these are for a single point along the flow? Where the flow depth is 50 m? Maybe a bonus figure before this showing where the cross section is taken might help?*

We improved the description.

*Table 1: Most of these are actually mainly referred to in the Appendix, I wonder if a subsection would be more beneficial here? Or move the whole table to the appendix where the bulk of the maths is anyhow?*

See previous point. Now table 2 is behind Fig.6, which has sample locations as per request of the referee. We think therefore is better in the main text

*Line 175: Is A13 the correct equation for flow temperature??*

Modified

*Line 183: by integration here – is this integrating over the 0 to 100%iles at 10m? Even then, 0.001 might be hard to get to?*

As specified, it is an integration over the flow thickness from 0 to 10 m above the ground level. We further clarified this.

*Line 184: extra space between of and the*

Modified

*Line 193: remove "value of"*

Done

*Line 194: What is section 14 of figure 2??*

Agree, deleted

*Line 201: flows occurred → flows that occurred*

Modified

*Line 222: DH → delta H*

Modified

*Lines 257 and 260: There has already been and equation (1) and (2)*

We modified all equations numbers

*Figure 5: This is quite hard to read, maybe remove the infrastructure data here or something?*

Agree, now it is Fig.6 and it has been modified accordingly

*Table 2: Suggest this goes to Appendix*

In our opinion, this table must stay in the main text also given the misunderstandings about the number of PDCs events taken into account. Furthermore, we modified the table's structure in order to further emphasize the number of units (PDCs deposits) taken into account in our analysis.

*Lines 298: As before – needs references / extra support for this statement*

Deleted

*Line 300: "not weaker" is a bit vague – what parameters are you talking about here?*

Modified

*Lines 309-310: Does this addition of zeros force the contours to not enter these areas? Where were the zeros added? Is this saying that future PDCs can never reach locations they have not been to before?*

We agree with the reviewer that the "zero" line may somehow force the contouring because the fact that the PDC deposit is not visible does not necessarily rule out the arrival of that PDC in that point, but this discrepancy may happen where the impact parameters are already very close to zero, as it can be seen from the label of the last contour. We added a statement clarifying this in lines 875-877.

*Figure 7: This is a great way of showing the difference in impact parameters across a map but I cannot make the leap to "hazard map". What is the statement you would make associated with a point, e.g., at Afragola airport – what is the hazard there? Is it not "Given an eruption in the next 50 years the exceedance probability….", it is not "In the next eruption we would expect…..", what is the hazard-based statement you can get from these?*

Please see our reply above. Also, see our new text at lines 848-853

*Line 322: differentiating → differentiate.*

Modified

*Line 380: "consider the PDCs of all eruptions" → nope, you've considered (and summed essentially) the 6 largest (assumed due to best-preserved) PDCs from previous eruptions.*

See our reply above concerning the number of PDCs considered. Anyway, we modified the sentence in lines 853-854.

*Line 379: exceeds → exceed*

Modified

*Lines 391-392: Highly speculative here, suggest removal.*

Agree.

*Lines 394-397: I don't think you did a "probabilistic approach that accounts for the variability of the intensity…"*

See our new text at lines n507-520 justifying the assumptions. Our approach is probabilistic in the sense that the outputs, which in our opinion indeed account for the variability of the PDCs intensity, are probability density functions (from which we extrapolated the impact parameters values at 16% exceedance probability) that account for input data (e.g., grainsize, density distributions) and modelling uncertainty. We are not making any assumption on the future scenario nor on the calculation of the probability of an eruption in 50 years. In our opinion, the sentence is still valid after the modifications made to clarify our approach thanks to your suggestion.

*Lines 408-410: Same big claim again – requires references.*

Changed at lines 1181-1182

*Line 434: "also the higher end of epistemic uncertainty." → no idea what this means here*

Changed at 1199-1200

*Line 478: extra ")" needs removing*

Modified

**Reviewer 2**

*I have made many editorial suggestions throughout the manuscript, as can be seen on the pdf mark-up I have uploaded. The authors should consider that many, probably most, readers will not be familiar with the details of hazards work at Vesuvius. The authors could do a better job of setting up the context of their work, for example by including a figure at the beginning that shows the current red zone. They could then clearly discuss the issue that the red zone map does not have any details of the impact parameters, i.e. one is either "in" or "out." Also an issue with the red zone approach is the assumption of a certain (or narrow range of) eruption scenarios, which does not take advantage of the range of behaviors that the volcano has exhibited, as recorded in the deposits. This sort of set up would help then when we come back around to the discussion and conclusions at the end. Also, it would help if some terms were defined when they are first used (see notes in ms).*

Now a new Fig.1 with the red zone has been added

*Here are my major comments (Note that the most important comments are related to the Appendix model):*

*Line 65 – is the value of 34% a media value? What is the uncertainty around that?*

It is the probability of an eruption at Vesuvius in the next 50 years (starting from low activity level) as reported by Selva et al. (2022) using their model. The probability comes from the curve probability vs. time extrapolated at 50 years. For further details, the readers can refer to the cited paper, which is open access.

*Figure 1 – see my comments. Improvements are needed.*

Done, now it is Fig. 2

*In general there seems to be a sort of narrow approach taken to interpreting the deposits, meaning that it is stated – as fact, with no consideration of other possibilities and very little observational motivation – that all massive deposits are sourced the impact zone process, and that each current's deposits reflect the bipartite model. I would like to see the authors discuss a broader range of possibilities for the origins of these deposits. I have made some notes on the ms that I hope are of use.*

Agree, added in section 2 a more detailed description of the different possible origin of deposits

*Line 115 and Figure 2 – what is the basis for the location and size of the zone of impact for the Mercato eruption? This is sort of presented as a fact, but what is the evidence? What was the topography of the volcano like at that time? Also, Fig 2 needs some "help" to make it easier for the reader.*

Added a sentence for justifying the location of the collapse zone

*Line 230 – here and other places layers are interpreted to be part of a single current, but no evidence is presented to convince the reader that this is the case. It would really help strengthen the paper if some summary of the evidence used to determine whether a sequence of layers are from a single current.*

Added some detail: the main reason is stratigraphic continuity, no erosional contact, similar occurrence in other deposits, added references

*Line 310 – there needs to be more information about how the hazard maps are produced. If I understand correctly, at each point where there are field data, the model was used to compute a pdf of the impact parameters. The 84th percentile value at that point was then used, along with the values at other points and from other eruptions, to make contours of individual impact parameters (such as dynamic pressure). If this is correct, then this should be said in a simple and straightforward way (I don't mind if the authors use the exact words above, if they make sense). Was there any sort of integrative method such as Monte Carlo used to sample across the uncertain parameters? It is critical that the process used to generate the maps, and the associated assumptions, be totally clear and transparent, if the authors want to maximize their use in decision making.*

We thank you and the other reviewer for giving us the chance to further clarify our approach. We applied PYFLOW to all the PDC outcrops that we could identify. At each outcrop we calculated the probability density function of the impact parameters, from which we extracted the value corresponding to the 84$^{th}$ percentile (16% exceedance probability). The model is publicly available, with a user manual that fully goes through the statistical treatment behind the calculation of the impact parameters' probability density function, which would be too complex to include even in the appendix of a journal like NHESS. To better illustrate the construction of maps, a few sentences have been added to section 4.

*APPENDIX A (MODEL) COMMENTS:*

*The assumptions and simplifications behind the theoretical approach need to be explicitly and completely stated. For example, the concentration and velocity profile models are based on data/models that were originally obtained/tested for steady, uniform flows of water and particles in gently sloping flumes. These conditions are a departure from pyroclastic currents (especially surges) with time-dependent sources, non-isothermal and (sometimes) compressible flow, and complex topography. I do not mean to say that the approach is not useful, but please be clear about the simplifications involved and how they might affect the results.*

The original model and later modification has been applied several times since 2008. We think some effort was made to make the assumptions and applicability of the model clear throughout all the applications of the model. However, we added some further details on this in lines 508-522 for explaining that the probabilistic approach takes into account the unsteadiness and flow fluctuations of the PDC current

*Equation A12, which is a significant part of the theoretical logic chain, is only appropriate for particles settling through a stationary column of fluid. I.e., not for particles settling for a flow, where the duration also is important in determining final layer thickness. This is something that cannot be glossed over. The authors need to present a compelling argument about why this simplified model is ok to use for the application at hand. Note that the time factor is subsequently, and separately, used to compute the duration of a flow via the aggradation rate. While this makes sense, it seems to make A12 to be an internally contradictory part of the overall model.*

See previous point

*Equation A17 - If I understand this correctly, there is no heat transfer between particles and gas phases. In other words, the decline in temperature with distance is soley due to the dilution due to entrained air, i.e., a mixing model. Is this correct? If so this needs to be clearly stated as a simimplification, along with some description of the implications of the simplification compared to a model that transfers heat between particles and gas.*

Added a sentence justifying the approximation. Yes, it is similar to a mixing model, which we believe being reasonable for the typical particle sizes and concentrations of the flows under analysis. The approach is similar to, e.g., Cerminara et al. (2016), who made a

dimensional analysis that can be applied to our case to. We added statements about this in lines 673-675.

---

## Referee Report (RR1)

**Review of: Long-term hazard pyroclastic density currents at Vesuvius (Southern Italy) with maps of impact parameters.**

This is the second time I have reviewed this paper, and it has drastically improved from the original submission and in my opinion, it is now ready for publication. I include below a list of typographical corrections/suggestions.

Line specific comments:

L35     affects → affect, and maybe "in the path of" rather than "caught unprotected by"?
L43     being the area surrounding the volcano highly → as the area surrounding the volcano is
L73     sometime → sometimes
Fig 1    A, B, and C are quite hard to see in black, and slightly confusing with a, b, c, and d subfigures, maybe white and 1, 2, and 3? And the locations of these sites would be good to see on Fig 2 a.
L98     not sure this makes sense: "a high sedimentation rate that dumped turbulence"
L104    Fig 1a doesn't show a break-in slope? Maybe move the figure ref earlier in the sentence?
L143    is attitude the right word here?
L147    Ref needs fixing
Fig 3    I found this figure a bit confusing to read at first, where along the flow is this? Y(m) is height through the flow, so is this the cross section at the start of the flow? Or at a random location? Or is it identical throughout? (hopefully not the last one). The increase in dynamic pressure for the 84$^{th}$ percentile also wasn't discussed sufficiently in the text (and looks interesting!)
L169    I don't follow this short sentence.
L192    As a consequence, also flow duration is expressed → As a consequence, flow duration is also expressed
L193    What is section 14 in figure 2?
L219    I get a range of 500 – 700 m (800 – 300 = 500)
L222    delta
L231    undercurrent of Mercato eruption → undercurrent of the Mercato eruption
L245    some of the refs need fixing
L251    Such a behaviour → such behaviour
L269    remove third "of"
L276    remove " ' " from deposits'
L279    remove "the"
Fig 6    Looks useful but it took a while to understand. Could you also add information on eruption duration and maybe volume and VEI? And highlight which of these is the one you talked us through in section 3? Also, it looks like the first PDC from each eruption is the biggest, are there any other interesting insights you could provide from these data?
L328    all eruptions → all past eruptions
L329    inferring → to infer
L432    "to prospect" is an unusual phrase

---

## Author Response (AR2)

Dear Editor,

We are sending the last revised version of the paper "Long-term hazard of pyroclastic density currents at Vesuvius (Southern Italy) with maps of impact parameters" by Dellino et al, submitted to NHESS.

All the suggestions and comments of the two reviewers were taken into consideration.

On behalf of the Authors

Sincerely

Pierfrancesco Dellino